bioengineering/computational biology

microcirculation, constrained constructive optimization, parallel computing, haemodynamics, renal vasculature

**Authors for correspondence:**
G. D. Maso Talou
e-mail: g.masotalou@auckland.ac.nz
P. J. Blanco
e-mail: pjblanco@lncc.br

# Parallel generation of extensive vascular networks with application to an archetypal human kidney model

L. F. M. Cury[1,4], G. D. Maso Talou[2],
M. Younes-Ibrahim[3,4] and P. J. Blanco[1,4]

[1]National Laboratory for Scientific Computing, LNCC/MCTI, Petrópolis, Brazil
[2]Auckland Bioengineering Institute, The University of Auckland, Auckland, New Zealand
[3]Faculty of Medical Sciences, Rio de Janeiro State University, UERJ, Rio de Janeiro, Brazil
[4]National Institute of Science and Technology in Medicine Assisted by Scientific Computing, INCT-MACC, Petrópolis, Brazil

GDMT, 0000-0002-5208-992X; PJB, 0000-0003-3527-619X

Given the relevance of the inextricable coupling between microcirculation and physiology, and the relation to organ function and disease progression, the construction of synthetic vascular networks for mathematical modelling and computer simulation is becoming an increasingly broad field of research. Building vascular networks that mimic *in vivo* morphometry is feasible through algorithms such as constrained constructive optimization (CCO) and variations. Nevertheless, these methods are limited by the maximum number of vessels to be generated due to the whole network update required at each vessel addition. In this work, we propose a CCO-based approach endowed with a domain decomposition strategy to concurrently create vascular networks. The performance of this approach is evaluated by analysing the agreement with the sequentially generated networks and studying the scalability when building vascular networks up to 200 000 vascular segments. Finally, we apply our method to vascularize a highly complex geometry corresponding to the cortex of a prototypical human kidney. The technique presented in this work enables the automatic generation of extensive vascular networks, removing the limitation from previous works. Thus, we can extend vascular networks (e.g. obtained from medical images) to pre-arteriolar level, yielding patient-specific whole-organ vascular models with an unprecedented level of detail.

# 1. Introduction

Nowadays, current multiscale modelling approaches and computational resources have stretched the boundaries of cardiovascular models and hemodynamic simulations. These *in silico* tools are starting to become part of medical research and play a major role in understanding the intricate physiological phenomena, developing novel therapeutic strategies, and improving diagnostic procedures [1–4].

Hemodynamics in the cardiovascular system (CVS) is a challenging problem involving blood flow dynamics and physiological phenomena occurring at different scales. For instance, the micro-circulation determines the peripheral hemodynamic environment, which affects the macro-circulation through the local and global regulation of peripheral resistance and compliance. In turn, the macro-circulation dictates the hemodynamic forces that promote the physiological interactions that unfold at the micro-circulation scale.

On the one side, complex three-dimensional models have gained momentum for the study of refined blood flow dynamics in localized regions of the CVS [5–7]. On the other side, one-dimensional models have been preferred to assess the hemodynamics at the scale of the entire CVS, ranging from large vessels to small arteries [8–10]. Most modelling approaches available in the literature employ simplified mathematical representations for the peripheral vascular beds [11–13]. In this sense, the over-simplification of the micro-circulation does not allow the study of sophisticated physiology at the level of smaller vascular structures, such as the size-dependent arterioles vasodilatory/remodelling responses. In turn, several works have studied the circulation at the scale of small arterioles and capillaries [14–17] employing lumped models (0D models: ordinary differential and algebraic equations), focusing on small pieces of tissue to analyse large vascular networks and investigating the relationship between blood flow, pressure, shear forces and mass exchange.

Medical images serve as input data to generate vascular domains. According to the modality, imaging techniques enable the visualization of a wide range of vascular vessels. Technological limitations do not allow us to analyse blood flow in the micro-circulation *in vivo*. Hemodynamics at the scale of arterioles and capillaries was investigated through mathematical models operating on top of vascular models constructed through high-resolution micro-tomographic images obtained from animal models [18,19]. In this context, automatic vascularization algorithms emerged more than two decades ago as a systematic approach to generate networks of interconnected vessels in regions of interest. These algorithms follow the hypothesis that network topology and geometry achieve an energetically efficient (or semi-efficient) perfusion of the vascularized tissues.

Several strategies have been devised to achieve that goal, from fractal approaches [20,21] to space-filling methods based on Voronoid tesselations [22,23] and cost function optimization methods [24–27]. The latter methods are termed the constrained constructive optimization (CCO) approach. Based on these CCO methods, several ramifications have been proposed to evolve in terms of model flexibility [28–30], enabling morphometric studies and blood flow simulations at the smaller scales in the CVS. Moreover, building *in vitro* vascular networks from scratch is being pursued by current technological advances [31–33]. Prior knowledge about network topologies suited to improve mass exchange and promote growth signalling is desirable for the optimized construction of these networks. In this experimental context, CCO-based methods can also provide insight into the advantage of certain topological configurations over others.

However, CCO-based approaches rely on a sequential optimization procedure in which points are sorted out, and a connectivity decision is made regarding the minimum of a cost function. These methods have been successfully employed to generate networks of up to 20 000 vascular segments. The main drawback in reaching larger numbers lies in the sequential approach and the ever-growing computational burden related to memory copying, neighbour searching, and network scaling when adding a new vessel. This has confined the possibilities of studying networks with a large number of vessels. Hence, modelling organ function accounting for the specific cellular physiology and its coupling with the systemic hemodynamic environment has remained an open challenge.

In light of the previous context, the main novelty of the present work is to develop a parallel strategy to efficiently generate extensive vascular networks using a CCO-based approach. Given a spatial domain to be vascularized, the first stage consists of creating a baseline network using a recently proposed aDaptive Constrained Constructive Optimization method (DCCO) [30]. At a second stage, we split the domain into non-overlapping subdomains, each of which is independently vascularized using the baseline network as the underlying vascular substrate. At a third stage, we consistently merge the networks. For further reference, we name the proposed parallel approach as PDCCO. The proposed PDCCO strategy exploits the multiscale nature of the CVS, for which the macro-scale circulation is

approached using a conventional CCO-based strategy, while the refined vascular networks at the micro-scale circulation are generated in a decoupled fashion, understanding the vascular generation problem as a local phenomenon.

Several numerical experiments are reported to investigate the capabilities and limitations of the proposed method. By comparing with sequentially generated networks (DCCO-networks), we study: (i) the impact of subdomain shapes in geometric and functional statistical indexes; (ii) the impact of the size of the baseline network in the resulting vascularization and (iii) the impact of the cost functional in the vascularization of a domain featuring multiple inlets. To illustrate the potential of the proposed method, we generate the arteriolar network in a piece of tissue resembling a portion of cortical renal tissue featuring 100 000 vascular segments. Finally, as an application towards the simulation of whole-organ function, we propose to create the arterial network in an archetypal model of the human kidney. Specifically, we construct the renal vasculature from scratch, starting at the renal artery until reaching the interlobular vessels, where afferent vessels are connected to glomeruli where filtration occurs. Specifically, we report the generation of an arterial tree in a highly non-convex geometry representing the renal cortex, featuring 100 000 terminal segments.

# 2. Material and methods

In this section, we briefly revisit the DCCO algorithm, and describe the proposed partitioned DCCO, termed PDCCO. Also, we describe the statistical analysis employed to compare vascular networks produced by PDCCO and DCCO.

## 2.1. Adaptive constrained constructive optimization

Let $T = \{v_i \,|\, i \in \{1, 2, 3, \ldots, N\}\}$ be a binary tree with $N$ rigid cylindrical vessels and $v_i = (r_i, x_i^p, x_i^d)$ be the $i$th vessel with radius $r_i > 0$, proximal position $x_i^p \in \mathbb{R}^3$ and distal position $x_i^d \in \mathbb{R}^3$. In turn, $\mathcal{V}$ and $\mathcal{T}$ are defined as the sets of admissible vessels and trees, respectively. Given a vessel cost functional $F : \mathcal{V} \to \mathbb{R}$, we define the total tree cost $\mathcal{F} : \mathcal{T} \to \mathbb{R}$ as

$$\mathcal{F}(T) = \sum_{v \in T} F(v). \tag{2.1}$$

Most commonly $\mathcal{F}$ measures total intravascular volume, i.e. $F(v) = \pi \,|\, x^d - x^p \,|\, r^2$. Nonetheless, we can change the optimality criterion by modifying $F$. For instance, to account for the energetic cost associated with angiogenesis [30], $F$ is defined as follows:

$$F(v) = c_v \frac{\pi |x^d - x^p| r^{\gamma - 1}}{V_{\text{ref}}^{\gamma/3}} + c_p \frac{r}{r_{\text{ref}}} + c_d \left( \frac{|x^d - x^p|}{l_{\text{ref}}} \right)^2, \tag{2.2}$$

where $V_{\text{ref}}$ and $l_{\text{ref}}$ are characteristic volume and length of the perfusion domain, $r_{\text{ref}}$ is a characteristic radius for the arterial tree, $c_v$, $c_p$ and $c_d$ are, respectively, the volumetric, proteolytic and diffusion non-negative coefficients such that $c_v + c_p + c_d = 1$. Parameter $\gamma$ characterizes the branching power law which governs the relation between parent and daughter vessels. Note that for $\gamma = 3$, we recover the classical volumetric cost functional, while for other values of gamma, the functional is consistent with the power law employed in the vascularization process.

In the CCO method, we search for $\overline{T}$ such that it minimizes a cost functional related to the genesis and maintenance of an arterial tree, namely

$$\overline{T} = \arg \min_{T \in \mathcal{T}} \mathcal{F}(T). \tag{2.3}$$

As initial condition, the method requires a initial tree $T_{\text{base}}$ or the position of an inflow point, radius and supplied flow rate, $x_0^p, r_0, Q_0$. Then, terminal vessels are sequentially added to the tree until it reaches a prescribed $N_{\text{term}}$ terminal segments. New segments are incorporated to the tree by picking a distal point inside the perfusion domain, say $x_{\text{new}}^d \in \Omega$ and by assessing potential connections to its neighbouring vessels, $B(x_{\text{new}}^d)$. For each vessel $v_p \in B(x_{\text{new}}^d)$, we take as possible bifurcation points, $x_{\text{bif}} \in \mathcal{B}(x_{\text{new}}^d, v_p)$, $n_{\text{bif}}$ points inside the triangle $\Delta x_p^p x_p^d x_{\text{new}}^d$ and check if $\overline{x_p^p x_{\text{bif}}}$, the modified parent vessel $v_p$, $x_{\text{bif}} x_p^d$, the sibling vessel $v_{\text{con}}$, and $\overline{x_{\text{bif}} x_p^d}$, the new terminal vessel $v_n$, are valid segments, i.e. the segments satisfy geometrical and hemodynamic constraints as detailed in [30]. If they are valid, we remove the original segment $\overline{x_p^p x_p^d}$, add the new ones to a provisional tree $T(x_{\text{new}}^d, x_{\text{bif}}, v_p)$ and

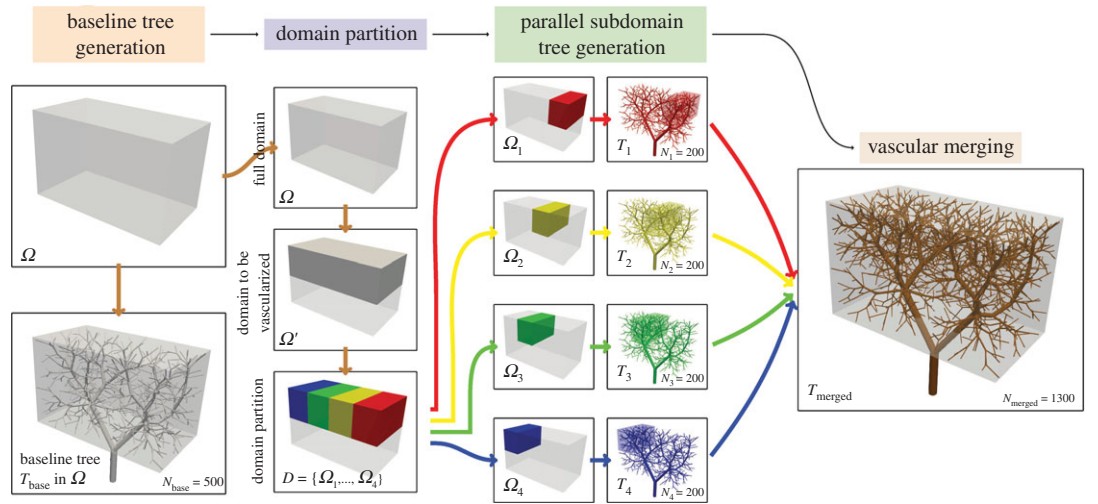

**Figure 1.** Parallel generation of vascular networks from a baseline tree. Example with $N_{part} = 4$, $N_{base} = 500$ and $N_i = 200$, $i = 1, 2, 3, 4$, resulting in $N_{merged} = 1300$.

evaluate $\delta\mathcal{F} = \mathcal{F}(T(x^d_{new}, x_{bif}, v_p)) - \mathcal{F}(T)$. If we cannot find any valid connections, we choose another random point and proceed again. Otherwise, we add to the enduring tree the vessels that resulted in the smallest value of $\delta\mathcal{F}$.

Along the optimization process that underlies the vascularization, it is necessary to solve compartmental fluid mechanics equations in the network. In doing this, conventional hypotheses are steady-state regime and Poiseuille blood flow. Since we are targeting the generation of massive networks, we consider the model of blood rheology that accounts for the Fàhræus–Lindqvist effect [34]. This implies viscosity depends on the vessel diameter, which introduces a nonlinearity in the vascularization process. We use fixed-point iterations as linearization procedure. This approach proved to be convergent and stable for all the experiments reported in this study. However, it is important to highlight that lack of convergence can be obtained for unrealistically small vessels (smaller than capillaries), for which the viscosity model is also questionable. For a detailed explanation of the DCCO algorithm, the reader is referred to [30].

The sequential optimization required by CCO and DCCO algorithms becomes increasingly computationally intensive as more vessels are added to the vascular tree. This aspect poses a limit to the utilization of these algorithms for the generation of extensive vascular networks, where a few recent works have studied how to alleviate such restrictions [18,35]. In the next section, we propose a divide and conquer approach for the optimization procedure to widen the applicability of this type of vascular tree generation algorithm.

## 2.2. Domain decomposition strategy

Let $\Omega$ be the perfusion domain with a region $\Omega' \subseteq \Omega$ concurrently vascularized and $\mathcal{D} = \{\Omega_1, \ldots, \Omega_{N_{part}}\}$ a partition of $\Omega'$. The domain decomposition strategy is fully described in algorithms 1 and 2, while figure 1 conceptually illustrates the major steps involved in the process. The proposed method follows the steps described below:

  (i)  Create a baseline tree, $T_{base}$, with $N_{base}$ terminal segments in $\Omega$. This baseline tree accounts for the global macro-scale features of the circulation.
  (ii)  For each $\Omega_i \in \mathcal{D}$, using $T_{base}$ as a starting point, add $N_i$ terminal segments in $\Omega_i$ to determine $T_i$ through the DCCO algorithm presented in §2.1. It is important that we consider only vessels $v \in T_{base}$ whose midpoint resides in $\Omega_i$. We call the set of such vessels $V_{part}$, as potential parents for the new segments in $T_i$. Thus, $v$ cannot bifurcate in more than one subdomain, guaranteeing the consistency of the merged tree.

  For each vessel $v$ added to $T_i$, a tuple $(x_{new}, x_{bif}, x^p_p, x^d_p)$ is inserted, in sequential order, into the list $\mathcal{L}_i$, as in line 28 of algorithm 1, indicating the distal and proximal coordinates of $v$ and its parent vessel. During the addition of a new vessel to a subdomain $\Omega_i$, only the terminal

blood flow in such regions is homogeneously redistributed, thus, terminal blood flow distribution outside $\Omega_i$ remains unchanged. This strategy yields a more consistent common tree across the parallel generated trees $T_i$. That is, we maintain $q_{out}$ fixed for terminal vessels not in $V_{part}$, and distribute the remaining flow $Q_{part}$ homogeneously for terminal vessels inside the domain. Such boundary conditions come into force on lines 22 and 30 of algorithm 1. This parallel stage accounts for the refined generation of micro-scale circulation networks, understood as a local phenomenon that does not significantly affect the macro-scale features of the circulation (see §3).

(iii) We initialize $T_{merged}$ as $T_{base}$ and, for each $\mathcal{L}_i$, we sequentially add the vessels ($x_{new}$, $x_{bif}$) whose parent is $(x_p^p, x_p^d)$ in the listed order. The radius of the added vessels are only scaled after the insertion of all tuples in the corresponding lists $\mathcal{L}_i$. So as not to search the whole tree for each insertion, we use a hash table whose keys are $x^p$ and $x^d$ to store the values $x_{new}$ and $x_{bif}$.

(iv) Once the tree $T_{merged}$ has $N_{merged} = N_{base} + N_1 + N_2 + N_3 + \ldots + N_{N_{part}}$ terminal segments, vessel radii are scaled according to the hemodynamic and geometric constrains as detailed in [30]. This is described in line 13 of algorithm 2. The merging process consists in coupling the macro- and micro-scales by providing consistency to the concurrent networks as a whole unit.

**Algorithm 1.** PDCCO part $\Omega_i$ vascularisation.

---

**Input**: Baseline tree $T_{base}$, subdomain $\Omega_i$ of $\mathcal{D}$ partition, the number of terminal segments to add $N_i$

**Output**: The subdomain tree $T_i$ with $N_{term} = N_{base} + N_i$ terminal segments.

1 $T_i \leftarrow T_{base}$
2 Let $V_{term}$ be the set of the terminal vessels of $T_i$
3 $q_{out} \leftarrow Q_{perf}/N_{base}$
4 $Q_{part} \leftarrow |V_{part} \cap V_{term}| \, q_{out}$
5 $N_{term} \leftarrow N_{base} + N_i$
6 **while** $|V_{term}| < N_{term}$ **do**
7 $n_{tries} \leftarrow 0$
8 Draw $x_{new}^d$
9 **while** $d(x_{new}^d, T_i) \geq l_{min}$ **do**
10 Draw $x_{new}^d$
11 $n_{tries} \leftarrow n_{tries} + 1$
12 **if** $n_{tries} \equiv 0 \pmod{N_{fails}}$ **then**
13 Update $l_{lim}$
14 List $\mathcal{L}$ // Empty
15 Optimal parent vessel $v_{p,opt}$ // Empty
16 Optimal bifurcation point $x_{bif,opt}$ // Empty
17 Minimal tree cost $\mathcal{F}_{min} \leftarrow \infty$
18 **foreach** $v_p \in B(x_{new}) \cap V_{part}$ **do**
19 **foreach** $x_{bif} \in \mathscr{B}(x_{new}^d, v_p)$ **do**
20 **if** the connection is valid **then**
21 $T' \leftarrow T_i(x_{new}^d, x_{bif}, v_p)$
22 Calculate $T'$ flow, radius, and resistance.
23 **if** $T'$ is a valid tree **then**
24 **if** $\mathcal{F}(T') < \mathcal{F}_{opt}$ **then**
25 $v_{p,opt} \leftarrow v_p$
26 $x_{bif,opt} \leftarrow x_{bif}$
27 **if** $\mathcal{F}_{opt} \neq \infty$ **then**
28 $\mathcal{L}$.push_back($x_{bif}, x_{new}^d, x^p, x^d$)
29 $T_i \leftarrow T_i(x_{new}^d, x_{bif}, v_p)$ // $|V_{term}| \leftarrow |V_{term}| + 1$
30 Calculate $T_i$ flow, radius, and resistance.
31 **return** $T, \mathcal{L}$

---

**Algorithm 2.** PDCCO merging.

**Input**: Baseline tree $T_{\text{base}}$ and the list sequence $(\mathcal{L}_i)_{i=1}^{N_{\text{part}}}$
**Output**: Merged tree $T_{\text{merged}}$

1  $T_{\text{merged}} \leftarrow T_{\text{base}}$
2  HashTable H
3  **foreach** $v \in T_{\text{base}}$ **do**
4  $\quad$ H.add$(v, x^p, x^d)$
5  **for** $i = 1$ **to** $N_{\text{part}}$ **do**
6  $\quad$ **foreach** $(x_{\text{bif}}, x_{\text{new}}^d, x^p, x^d)$ in $\mathcal{L}_i$ **do**
7  $\quad\quad$ $v_p \leftarrow$ H.at$(x^p, x^d)$
8  $\quad\quad$ H.remove$(x^p, x^d)$
9  $\quad\quad$ $T_{\text{merged}} \leftarrow T_{\text{merged}}(x_{\text{new}}^d, x_{\text{bif}}, v_p)$
10 $\quad\quad$ H.add$(v_n, x_{\text{bif}}, x_{\text{new}}^d)$
11 $\quad\quad$ H.add$(v_{\text{con}}, x_{\text{bif}}, x^d)$
12 $\quad\quad$ H.add$(v_p, x^p, x_{\text{bif}})$
13 Calculate $T_{\text{merged}}$ flow, radius, and resistance.
14 **return** $T_{\text{merged}}$

The simultaneous determination of the optimal points for each subdomain $\Omega_i$, as explained in Step 2 (see also figure 1), significantly reduces the computational burden, which amounts to the most computationally intensive stage in the sequential algorithm.

## 2.3. Data analysis

Vascular networks constructed using the PDCCO approach are compared to a vascular network built using the sequential DCCO approach. Given the random strategy to sample potential points, several instances are created for each case to properly characterize the stochasticity of the process and perform a fairer comparison between PDCCO and DCCO.

From the set of instances, either in the sequentially constructed network (DCCO network) or in the parallel approach (PDCCO network), we compare morphometric and functional measures, specifically vessel radii, length, aspect ratio and pressure, as a function of the vessel generation. Pressure is computed through Poiseuille Law, consistently with the underlying physics in the model. We place the reference (null) pressure at the inlet of the network. In doing this, we gather, across the different instances all vessels of a certain generation and build the corresponding box-plot. We report profiles of these quantities as a function of vessel generation. Also, we compare the distribution of vessels in the length-radii space.

As further verification of the vascular tree optimality resulting from the PDCCO algorithm, we report the discrepancy between the intravascular volume in PDCCO-networks and the mean volume of all DCCO networks. This discrepancy is normalized by the mean DCCO network intravascular volume and is termed relative volume error.

For the prototypical model of the kidney vascular network, we compute the Strahler order of each vascular segment to provide an alternative characterization of the radius, pressure and flow rate in the network. From vessel radii, we compute the total cross-sectional lumen area of all vessels combined, within each Strahler order bin, and the intravascular volume of the subtended tree associated with each vessel. Lastly, we construct a connectivity matrix, as a function of the Strahler order [36].

## 3. Results

In this section, we first evaluate the performance of the parallel approach (PDCCO) compared with the sequential approach (DCCO) in three idealized experiments, and later we analyse the performance of PDCCO to generate complex trees with a large number of vessels. Unless stated otherwise, we make use of the volumetric cost functional to generate the PDCCO and DCCO vascular networks.

The first experiment investigates the sensitivity in the resulting tree to the number of segments $N_{\text{base}}$ present in the baseline tree, from which the parallel vascularization in PDCCO is initiated. The second experiment refers to the effect of the domain shape, particularly the fact that partitions can become

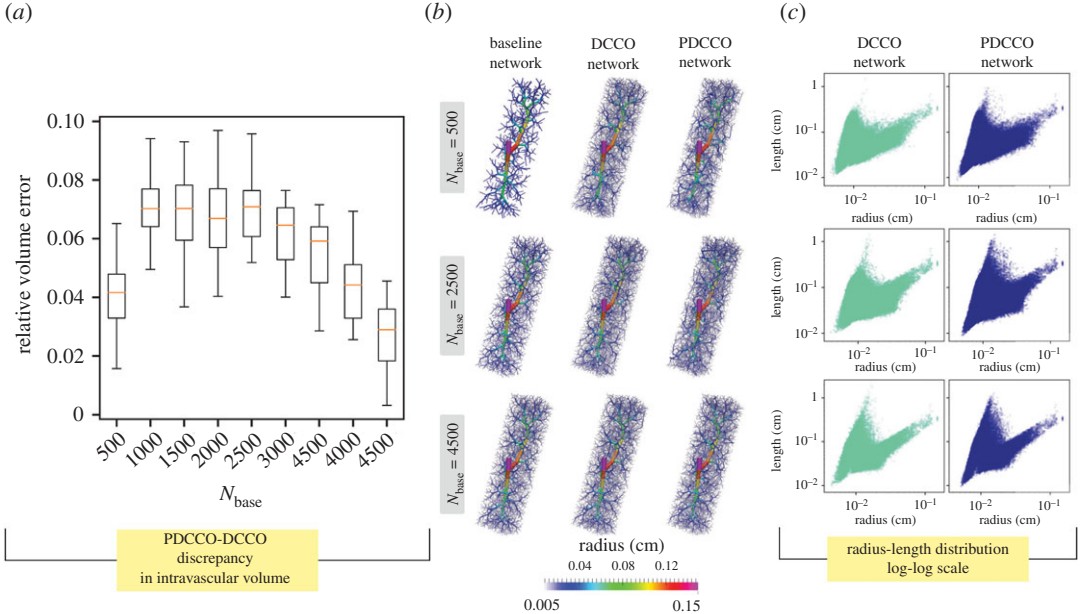

**Figure 2.** (*a*) Discrepancy of intravascular volume in PDCCO-generated networks ($n = 10$ per box) compared with the mean intravascular volume of all DCCO-generated networks ($n = 10$ per box) as a function of the number of segments ($N_{\text{base}}$) in the baseline network. The volume difference is normalized with the mean value of DCCO-generated networks for three values of $N_{\text{base}}$. (*b*) Instances of the baseline, DCCO and PDCCO networks, coloured with radius value. (*c*) Radius-length distribution for three values of $N_{\text{base}}$.

geometrically degenerated, in the resulting partitioned (parallel) vascularization. For these two experiments, the baseline network is shared by both approaches (DCCO and PDCCO). Therefore, the sequential approach continues the construction of the baseline network until reaching the total number of terminals, while the parallel approach uses the baseline network to initialize the concurrent vascularization of the different subdomains. In the third experiment, we study the impact of the cost functional definition in the resulting vascular network. Specifically, we propose to vascularize a domain with multiple inlets, and investigate the lack of sub-tree balance, in terms of flow rate carried by each inlet, as a function of the parameters that define the cost functional.

Regarding the PDCCO performance generating complex trees, we choose to study the scalability of the proposed approach in a prototypical portion of tissue with a parallelepipedal shape. The goal is to generate a vascular network in which over 100 K vascular segments are placed. Finally, an example of application in which we vascularize a renal cortex structure demonstrates the potential utilization of the proposed approach in a challenging scenario.

## 3.1. Sensitivity to the baseline tree

To analyse how the number of terminal segments in the baseline tree ($N_{\text{base}}$) affects the final network, we executed nine sets of simulation scenarios, each set with a different $N_{\text{base}}$ value. All executions in a set use the same parameters except for the random generation seed used for the point generation (see Algorithm 1, lines 8 and 10). Each set has 10 instances of vascular networks with 5000 terminal vessels generated by the PDCCO approach, aiming to characterize the stochasticity in the generation process due to the point generation. The domain geometry is defined as the parallelepipedal region from figure 1 divided into $N_{\text{part}} = 4$ subdomains. The number of terminal segments in the baseline tree for each set is given by $N_{\text{base}} \in \{500, 1000, 1500, 2000, 2500, 3000, 3500, 4000, 4500\}$. Also, we built ten instances of trees using the sequential DCCO approach to use as baseline for comparison.

In figure 2, the relative volume error is reported (*a*) as a function of the number of segments in the baseline network ($N_{\text{base}}$). Recall that this error measures the normalized discrepancy between the intravascular volume in PDCCO-generated networks and the mean of DCCO-generated networks. Observe that the median error approximates a plateau for $N_{\text{base}} \in \{500, 1000, 1500, 2000, 2500\}$, and then consistently declines as the number of elements in the baseline network is incremented. As expected, PDCCO renders a suboptimal solution for the optimization problem due to the decoupling

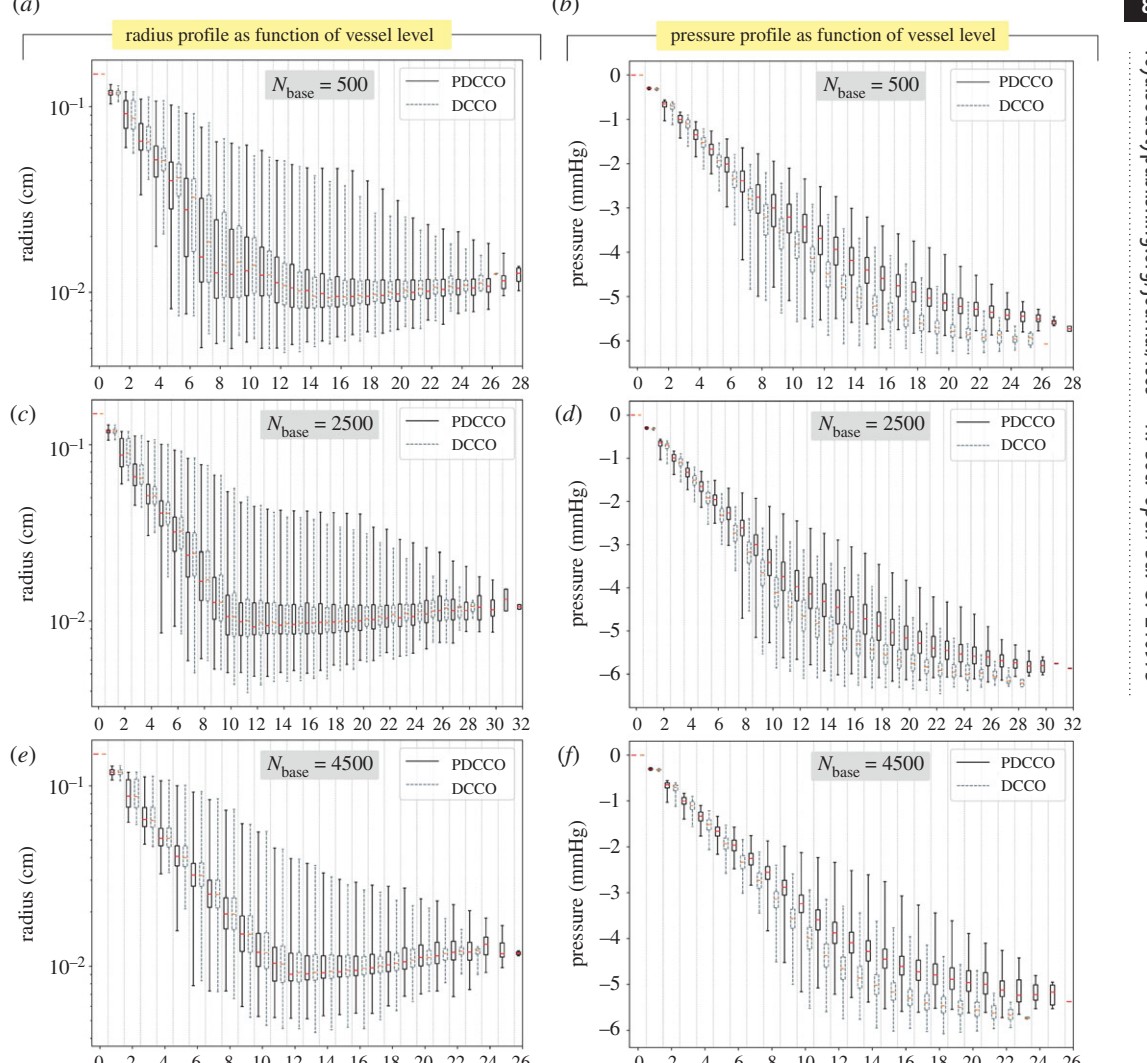

**Figure 3.** Statistical profile of vessel radius (a,c,e) and blood pressure (b,d,f) as a function of the vessel level in the vascular network. Results are reported for three selected cases of number of segments in the baseline network ($N_{base} \in \{500, 2500, 4500\}$).

of the problem. Nonetheless, the median of the disagreement with respect to DCCO is below 8% in the luminal volume of the generated tree. A complementary comparison between PDCCO and DCCO generated networks is given in the right panel of the same figure, where the distribution in the feature space defined by the vessel length and radius is shown. Observe that both PDCCO networks and DCCO networks share the same structural features in terms of vessel radius and length. Moreover, this common behaviour is independent from $N_{base}$. This piece of evidence suggests that the morphometric features of the vascular networks are invariant.

Figure 3 displays the statistical behaviour of the vessel radius and blood pressure depending upon the vessel generation (i.e. the number of bifurcations from the root vessel, $v_0$) in the vascular network. For each vessel generation, the box-plots given by the PDCCO-networks and by the DCCO-networks are compared head-to-head for three selected numbers of segments in the baseline tree $N_{base} \in \{500, 2500, 4500\}$. It can be observed that the radius interquartile ranges are in agreement between both PDCCO and DCCO algorithms along with the network depth. Similarly, whiskers are also in agreement, with subtle shifts in the middle part for $N_{base} = 2500$. In turn, the pressure has a very compact distribution in the first generations, and then the distribution widens. This feature is shared by both DCCO and PDCCO algorithms regardless of $N_{base}$. In more detail, we can observe that the pressure drifts upwards after the 6th generation in the PDCCO networks, compared to the DCCO networks. In other words, the pressure drop as predicted by PDCCO networks is slightly smaller than the pressure predicted by DCCO networks.

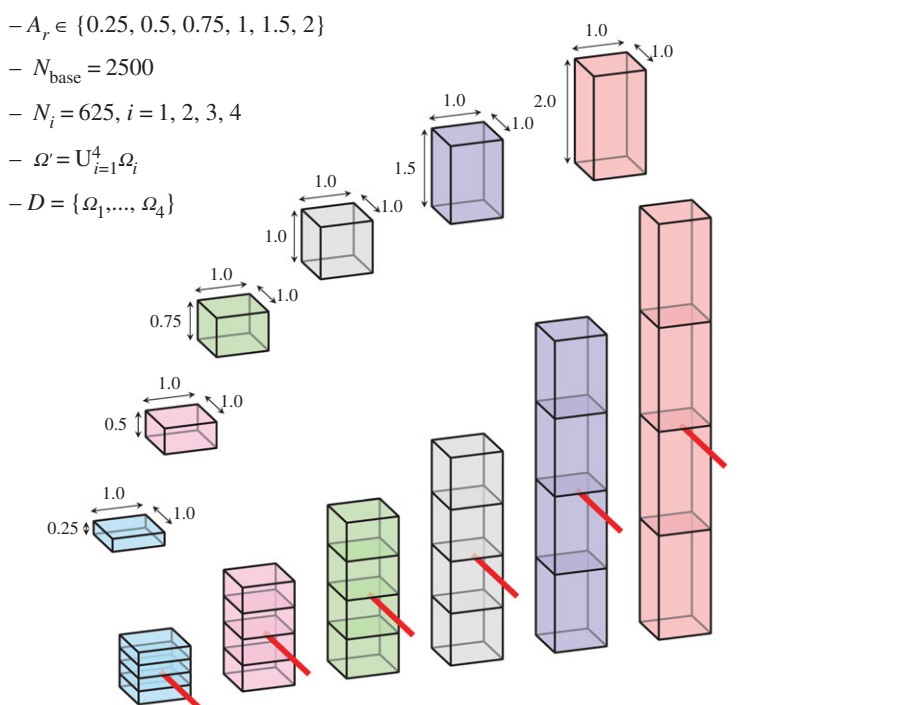

**Figure 4.** Sensitivity of parallel vascularization to domain aspect ratio. Example with $N_{part} = 4$, $N_{base} = 2500$ and $N_i = 625$, $i = 1, 2,$ 3, 4, resulting in $N_{merged} = 5000$.

## 3.2. Sensitivity to the domain shape

We have designed an experiment to check whether the subdomain's shape functionally affects the generated vascular network. As with the first test, we ran 10 instances for each setup, all with the same parameters except for the random number generator seed. In total, we have six different setups where PDCCO and equivalent DCCO vascular networks are generated. For the PDCCO trees, we had $N_{part} = 4$ and $N_{base} = 2500$ with the final tree having 5000 terminal segments. To control the effects of changing the domain volume, we scaled the influx and root radius according to the allometric laws $Q \propto V$ and $r_0 \propto V^{3/8}$ [37]. The domain is defined by a cuboid of dimensions $4L \times 1 \times 1$ with four cuboid subdomains of dimensions $L \times 1 \times 1$ adjacently disposed on the vertical axis, as shown in figure 4. The influx position is taken as $(L/10)$. We choose $L \in \{0.25, 0.5, 0.75, 1.0, 1.5, 2.0\}$ so as to have the subdomain aspect ratio $A_r \in \{0.25, 0.50, 0.75, 1.00, 1.50, 2.00\}$ for our six different setups. Figure 5*a* displays the relative volume error (discrepancy between the intravascular volume in PDCCO-generated networks and the mean of DCCO-generated networks). Observe that the median error for the different values of $A_r$ (aspect ratio) remains bounded, below 8%. It is worthwhile to note that the error is smaller for more degenerated subdomains ($A_r \in \{0.25, 0.50\}$), grows and then stagnates for $A_r \in \{1.00, 1.50, 2.00\}$. In figure 5*b*, we illustrate an example of the baseline network, the DCCO network and the PDCCO network for different values of $A_r$. In figure 5*c*, the radius-length distribution is shown for the DCCO and PDCCO networks. Again, in this case, the PDCCO algorithm manages to deliver networks structurally similar to the DCCO algorithm. Also, note that by modifying $A_r$, the change in the distribution shape is minimal. That is, the PDCCO algorithm emulates the overall morphometric behaviour of the DCCO network.

In figure 6, vessel radius and blood pressure distributions are reported as a function of the vessel generation. The box-plots for each generation from the generated PDCCO networks and DCCO networks are compared for three selected aspect ratios $A_r \in \{0.25, 1.00, 2.00\}$. Radius interquartile range in PDCCO-generated networks closely follows the behaviour of the networks generated by the DCCO algorithm. Only subtle discrepancies are noticed at most distal locations, where the number of vessels per box drops significantly. Whiskers, in turn, are similar in the first generations and start to deviate as we go over the 10th generation. That behaviour is independent of $A_r$. Due to the highly nonlinear relationship between pressure and vessel radius, the subtle differences observed in the left column plots become more evident in the right column plots. Remarkably, the

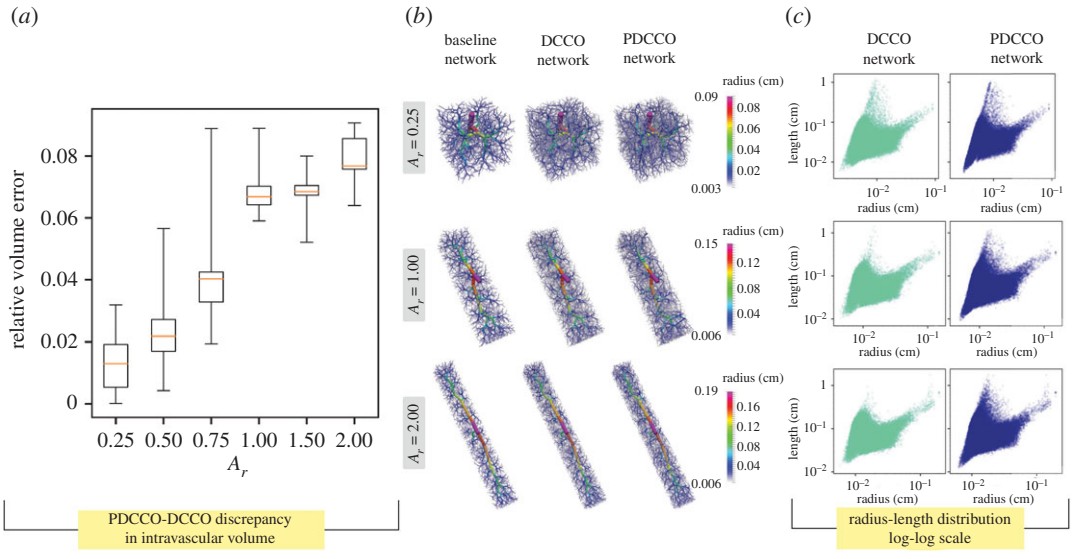

**Figure 5.** (*a*) Discrepancy of intravascular volume in PDCCO-generated networks compared with the mean intravascular volume of all DCCO-generated networks as a function of subdomain aspect ratio ($A_r$). The volume difference is normalized with the mean value of DCCO-generated networks. (*b*) Instances of the baseline, DCCO and PDCCO networks, coloured with radius value. (*c*) Radius-length distribution for three values of $A_r$.

**Figure 6.** Statistical profile of vessel radius (*a,c,e*) and blood pressure (*b,d,f*) as a function of the vessel level in the vascular network. Results are reported for three selected cases of subdomain aspect ratio ($A_r \in \{0.25, 1.00, 2.00\}$).

**Figure 7.** Partitioned vascularization of cubic domain with four inlets for different cost functional parameters ($c_v$, $c_p$, $c_d$). Top panel shows the networks which achieve 'nearly equal' and 'largely unequal' flow rate balance among the four inlet branches (% of flow rate informed next to each inlet). Bottom plots feature the radius and pressure profiles as a function of the vessel level.

overall distribution shape of the pressure profile is in agreement in PDCCO networks and in DCCO networks for the different values of $A_r$. A close-up analysis shows us that for $A_r = 0.25$, the pressure range is greater in PDCCO networks for a given vessel generation, and the median remains slightly above that in DCCO networks. For larger values of $A_r$, the pressure range drifts upwards, including the whole interquartile range, in PDCCO networks. Hence, the pressure drop is slightly smaller in PDCCO networks than in DCCO networks.

## 3.3. Domain with multiple inlets

In this section, we build a vascular network in a domain supplied by multiple inlets. We consider a cubic domain centred at the origin (side length of 2) and an external network supplying the domain through four opposed planes (figure 7). The tuple of parameters ($c_v$, $c_p$, $c_d$) that characterize the cost functional (see expression (2.2)) is defined by taking steps of 0.1 for all the parameters, from 0 to 1 for each parameter, with the constraint that $c_v + c_p + c_d = 1$. Hence, we build a total of 55 baseline tree vascular networks, and investigate the flow rate distribution among the four major supply branches. Out of these 55 baseline trees, we analyse two exemplar cases: (i) a close to an 'equal' flow distribution; and (ii) an 'unequal'

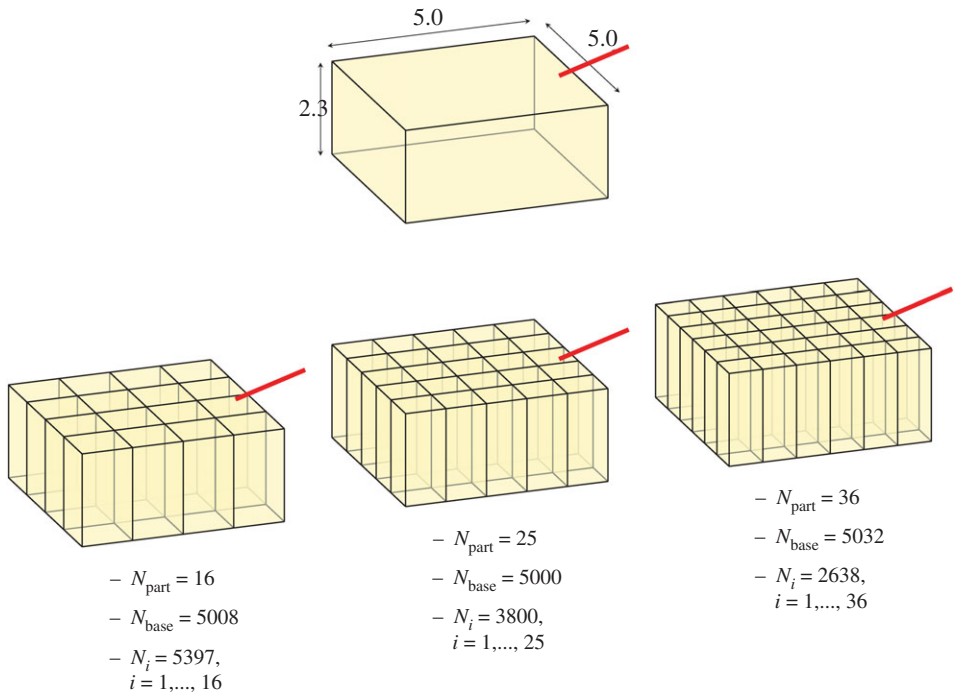

**Figure 8.** Parallelepipedal region (domain dimensions $5 \times 5 \times 2.3$) vascularized with 100 000 terminal vessels. The domain is partitioned into different number of parts, $N_{part} \in \{16, 25, 36\}$. The number of vessels in the baseline tree is roughly $N_{base} = 5000$.

flow distribution. For each exemplar, we continue the vascular generation with the PDCCO approach performing the partitioned vascularization stage and the merging stage.

The baseline tree is built with $N_{base} = 200$, and the domain is partitioned into eight octants (side length of 1) for which $N_i = 400$, $i = 1, \ldots, 8$ vessels are added, yielding a final network of $N_{merged} = 3400$ vessels.

Figure 7 displays the vascularized domain and the partitions. The vascular networks featured in the figure illustrate the two opposite scenarios concerning the flow rate balance among the four inlets. The scenario termed 'equal' is the one for which the flow rate standard deviation in the four branches achieves the smallest value. This leads to inlet branches with nearly equal size. The scenario called 'unequal' is the one for which the flow rate standard deviation in the four branches reaches the highest value, resulting in inlet branches with unequal size, and networks whose extension varies substantially. Next to each inlet branch, we inform the corresponding flow rate fraction. We also report in the same figure the radius and pressure distributions for these two extreme cases.

Macro-scale features of the vascular network are determined in the generation of the baseline network, which is accomplished using the sequential DCCO algorithm. This structure depends on the cost functional parameters as expected. The PDCCO approach initiates the process with an already balanced/unbalanced tree, and it continues to grow within the domain in a parallel fashion, regardless of the extension of each vascular subnetwork and flow rate balance. This is confirmed by the subtle changes in the flow distribution when moving from the baseline tree $T_{base}$ to the merged one $T_{merged}$. Hence, the partitioned stage is responsible for the definition of refined vascular features encountered at smaller scales. Indeed, as also shown in figure 7, the distribution of vessel radius and pressure as a function of the vessel level changes substantially in the first levels, although higher levels present similar values.

## 3.4. Scalability

In this section, we design an experiment to test the scalability of the parallel approach to generate a vast network. This allows us to study the computational gain obtained with the proposed strategy. Let us consider the parallelepipedal region from figure 8. The domain is partitioned into 16, 25 and 36 equally sized subdomains, and the number of terminal vessels in the baseline network is $N_{base} \approx 5000$—this is to keep the remaining $100\,000 - N_{base}$ vessels as a number divisible by $N_{part}$. First, we

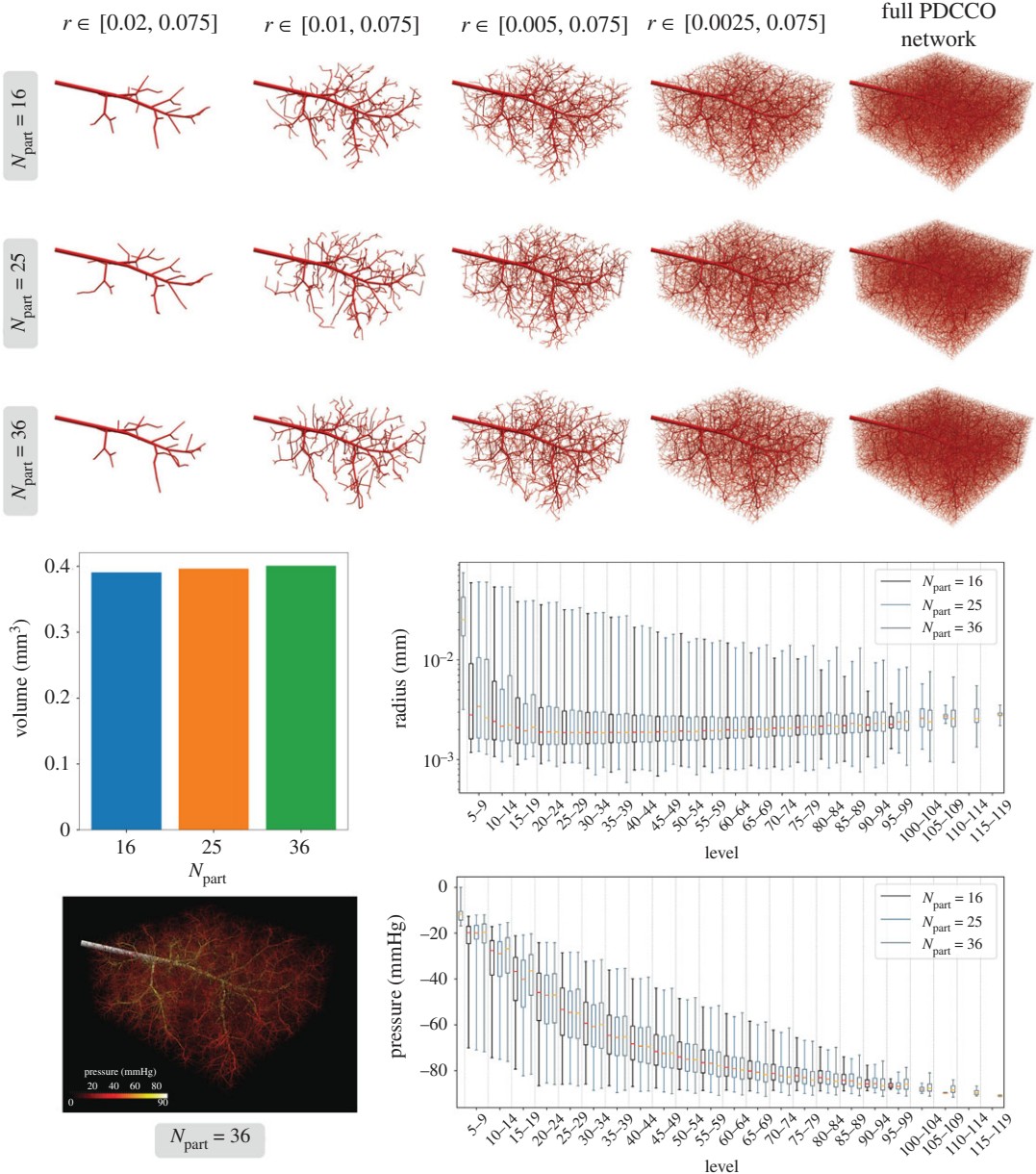

**Figure 9.** Vascularization of a parallelepipedal region employing different number of subdomains ($N_{part} \in \{16, 25, 36\}$). The baseline network contains 5000, and the final number of terminal segments in the merged network is 100 000. Radius and pressure profiles, and intravascular volumes for the different networks are generated. The resulting pressure distribution in the network is also shown.

generate the baseline tree in the entire domain, and then the parallel generation is executed, after which the $N_{part}$ networks are merged to deliver the final network. The flow rate at the inlet of the network is set to be $Q_{in} = 2$ mm$^3$ s$^{-1}$ and the inlet radius is $r_1 = 0.075$ cm.

Figure 9 displays the resulting vascular networks for the different partitions employed. Over the top of the figure, by thresholding the network with the vessel radius, it is possible to see that the dominant vascular structures remain unchanged when altering the partitioning and the parallel generation of the vascular tree. This outcome is fundamental to confirm that the parallel version of the DCCO algorithm does not introduce artefacts in the resulting network. The equivalence between the sequential and the parallel algorithms can also be verified in the radius and pressure profiles reported at the bottom of the figure. The box-plot for the three different partitions are compared alongside as a function of vessel generation. Both geometrically and functionally, the three networks behave equivalently. Over the left panel in figure 9, the intravascular volumes of the different networks are compared, where no

**Table 1.** Time (in hours) spent in constructing the baseline tree and in the vascularization of each subdomain (mean and s.d.) until reaching 100 000 terminal vascular segments.

| $N_{part}$ | time in network construction (h) | |
| --- | --- | --- |
| | baseline | subdomains (avg ± s.d.) |
| 16 | 3.27 | 5.53 ± 0.58 |
| 25 | 3.27 | 3.00 ± 0.36 |
| 36 | 3.27 | 2.07 ± 0.38 |
| $N_{seq}$ | time* in sequential construction (h) | |
| 95 000 | 2230 | |

*estimated time using a projection from sequentially constructed vascular networks.

**Table 2.** Anatomical data used as input for the vascularization of the prototypical model of the human kidney.

| data | reported value | model |
| --- | --- | --- |
| dimensions (cm) | $11 \times 5 \times 2.5$ [40] | $11.7 \times 6.54 \times 4.92$ |
| total renal volume (cm³) | 152.9 ± 7.5 [38] | 155.32 |
| cortical volume (cm³) | 72.5 ± 2.5 [38] | 74.65 |
| medular volume (cm³) | 80.4 ± 2.7 [38] | 80.65 |
| cortical/medular volume ratio | 0.92 ± 0.03 [38] | 0.93 |
| number of renal pyramids | 8–18 [41] | 13 |

significant differences appear. Finally, at the bottom left panel in the same figure, we report the pressure distribution in the network generated with $N_{part} = 36$ subdomains.

For this experiment, we measure the cost in terms of time spent to address the different phases in the algorithm. Table 1 reports the time spent (in hours) to build the baseline network ($N_{base} = 5000$), as well as the time spent in the construction of the vascular networks for each subdomain (mean and s.d.). Note that, when removing the shared baseline network, the PDCCO algorithm scales linearly concerning $N_{part}$ as the subdomain vascularization problems are completely decoupled.

To compare the estimated cost in building such a network using the sequential DCCO algorithm, starting from the baseline network, we executed the sequential algorithm for different numbers of terminal segments, specifically $\{1, 2, 3, 4, 5, 6, 7, 8, 9, 10, 15, 20, 25, 30, 35\} \cdot 10^3$ terminals. Regression analysis yielded a cost of the order of $N_{seq}^3$, wirh $N_{seq}$ being the number of terminal segments added to the baseline network sequentially. Then, we estimated the cost of sequentially adding $N_{seq} = 95\,000$ terminals to the baseline network. This estimated cost is also reported in table 1.

## 3.5. Renal vascular network

In this last experiment, the goal is to illustrate the application of the PDCCO method in a complex scenario: the vascularization of the renal cortex structure. To this end, we have created a prototypical geometry of a left human kidney from reported anatomical data [38]. For a body surface area of $1.65\,m^2$ [8,39] and from the data reported in [38], we determined total, cortical and medullary volumes for our model. Then, we scaled the geometric model until achieving appropriate dimensions. Morphometric data for the kidney model is detailed in table 2. From the point of view of the PDCCO algorithm, this is the input data that characterizes the domains to be vascularized.

At the beginning of the renal vasculature, the renal artery bifurcates into many segmental vessels, that continue to branch into interlobar arteries that extend through the renal columns surrounding the renal pyramids, to finally give rise to the arcuate arteries. Arcuate arteries branch into interlobular vessels. The renal cortex is one of the most densely vascularized regions in the human body. Within 72 cm³, up to 1

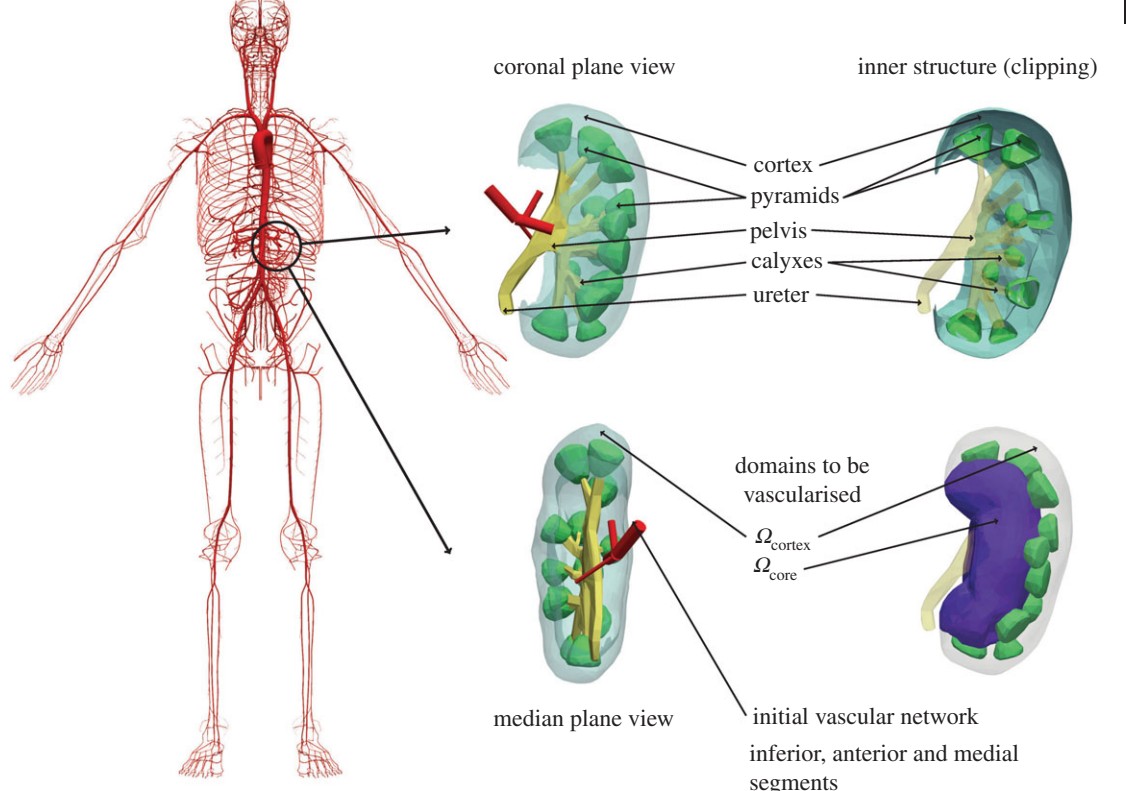

**Figure 10.** Geometric model of the human kidney and domains to be vascularized using in a multi-stage PDCCO strategy.

million glomeruli can be found. Glomeruli are connected to tubules, forming the nephrons, which are the basic functional units responsible for blood filtration. Each glomerulus is a capillary bed contained in Bowman's capsule, where the blood crossing the glomerular membrane filtration turns into the so-called filtrate. An afferent arteriole supplies each glomerulus (diameter in the order of 25 µm), which in turn branches off from the aforementioned interlobular arteries (diameter in the order of 50–100 µm). Such a compact vasculature is guested in a spatial domain containing many obstacles. The set formed by renal pelvis, calyxes and pyramids poses intrinsic spatial restrictions from early embryonic stages that constrain the development of the vascular network.

The initial vascular model, the main components of renal anatomy, and the initial vascular network, from which the proposed PDCCO algorithm is applied, are displayed in figure 10. In this experiment, the goal was to create a vascular network of the entire kidney. The kidney vascularization was achieved through a multi-stage strategy as described in [30], and detailed in table 3.

All the main ingredients involved in the process are schematically presented in figure 10. First, an initial renal vascular network containing the three major arterial vessels, inferior, anterior and medial segmental arteries (see the three-vessel network in figure 10), was handcrafted from the renal artery [8,39]. Then, we triggered the first vascularization stage from these three major segmental vessels (network $T_0$) using a sprouting cost functional with angle constraint and spanning the medulla domain, called $\Omega_{core}$ (see purple region in figure 10). The second and third stages are responsible for placing 5000 terminal segments in the $\Omega_{cortex}$ region (see region which surrounds $\Omega_{core}$ in figure 10). These stages, and the ones that follow, optimize a volumetric cost functional to drive the vascularization. The difference between these two stages lies in the size of the neighbourhood (determined by $f_n$), where potential connections between the new terminal point and each vessel are sought. The outcome of these two stages is the vascular network $T_3$, which is used as the baseline network to launch the parallel partitioning proposed in this work. The kidney domain is divided into eight parallelepipedal subdomains placed at different transverse plane locations (see vascularization of $\Omega_i$, $i = 1, \ldots, 8$ in figure 11). Four stages with progressive reduction in the neighbouring factor $f_n$ were used to vascularize the $\Omega_{cortex}$ domain, until reaching a vascular network featuring 100 000 terminal vessels.

Concerning the setting of the PDCCO algorithm, the baseline network (called $T_3$ in table 3) contains $N_{base} = 5000$ terminal vessels. As said above, eight partitions were created by slicing the kidney with

**Table 3.** PDCCO parameter setup for kidney vascularization. Coefficients and reference values for $\mathcal{F}_{sprout,1}$ are $c_v = 0.999$, $c_p = 0.0$, $c_d = 0.001$, $V_{ref} = 100$ cm$^3$ and $r_{ref} = 1.0$ cm. $Q$ and $R$ are respectively the quotient and remainder of the integer division of $N_i$, $i \in \{1, 2, 3, \ldots, 8\}$ by 4.

| stage | domain | $\mathcal{P}_{geo}$ $(\gamma, \delta)$ | $\mathcal{P}_{opt}$ $(v, f_r, f_n, \Delta v)$ | initial condition | N | cost functional | vessel type | angle constraint $(\theta_{min}, \phi_{min})$ | branching type |
|---|---|---|---|---|---|---|---|---|---|
| common base | | | | | | | | | |
| $\mathcal{S}_1$ | $\Omega_{core}$ | (3, 0) | (1.0, 0.9, 9.0, 21) | $T_0$ | 21 | $\mathcal{F}_{sprout,1}$ | distribution | (30°, 0.0) | distal |
| baseline | | | | | | | | | |
| $\mathcal{S}_2$ | $\Omega_{cortex}$ | (3, $\delta_1$) | (1.0, 0.9, 8.0, 7) | $T_1$ | 1666 | $\mathcal{F}_{vol}$ | distribution | (30°, 0.0) | versatile |
| $\mathcal{S}_3$ | $\Omega_{cortex}$ | (3, $\delta_1$) | (1.0, 0.9, 4.0, 7) | $T_2$ | 3334 | $\mathcal{F}_{vol}$ | distribution | (30°, 0.0) | versatile |
| partitioned | | | | | | | | | |
| $\mathcal{S}_4$ | $\Omega_{cortex}$ | (3, $\delta_1$) | (1.0, 0.9, 1.0, 7) | $T_3$ | $Q$ | $\mathcal{F}_{vol}$ | distribution | (30°, 0.0) | versatile |
| $\mathcal{S}_5$ | $\Omega_{cortex}$ | (3, $\delta_1$) | (1.0, 0.9, 0.5, 7) | $T_4$ | $2Q$ | $\mathcal{F}_{vol}$ | distribution | (30°, 0.0) | versatile |
| $\mathcal{S}_6$ | $\Omega_{cortex}$ | (3, $\delta_1$) | (1.0, 0.9, 0.25, 7) | $T_5$ | $3Q$ | $\mathcal{F}_{vol}$ | distribution | (30°, 0.0) | versatile |
| $\mathcal{S}_7$ | $\Omega_{cortex}$ | (3, $\delta_1$) | (1.0, 0.9, 0.125, 7) | $T_6$ | $4Q + R$ | $\mathcal{F}_{vol}$ | distribution | (30°, 0.0) | versatile |

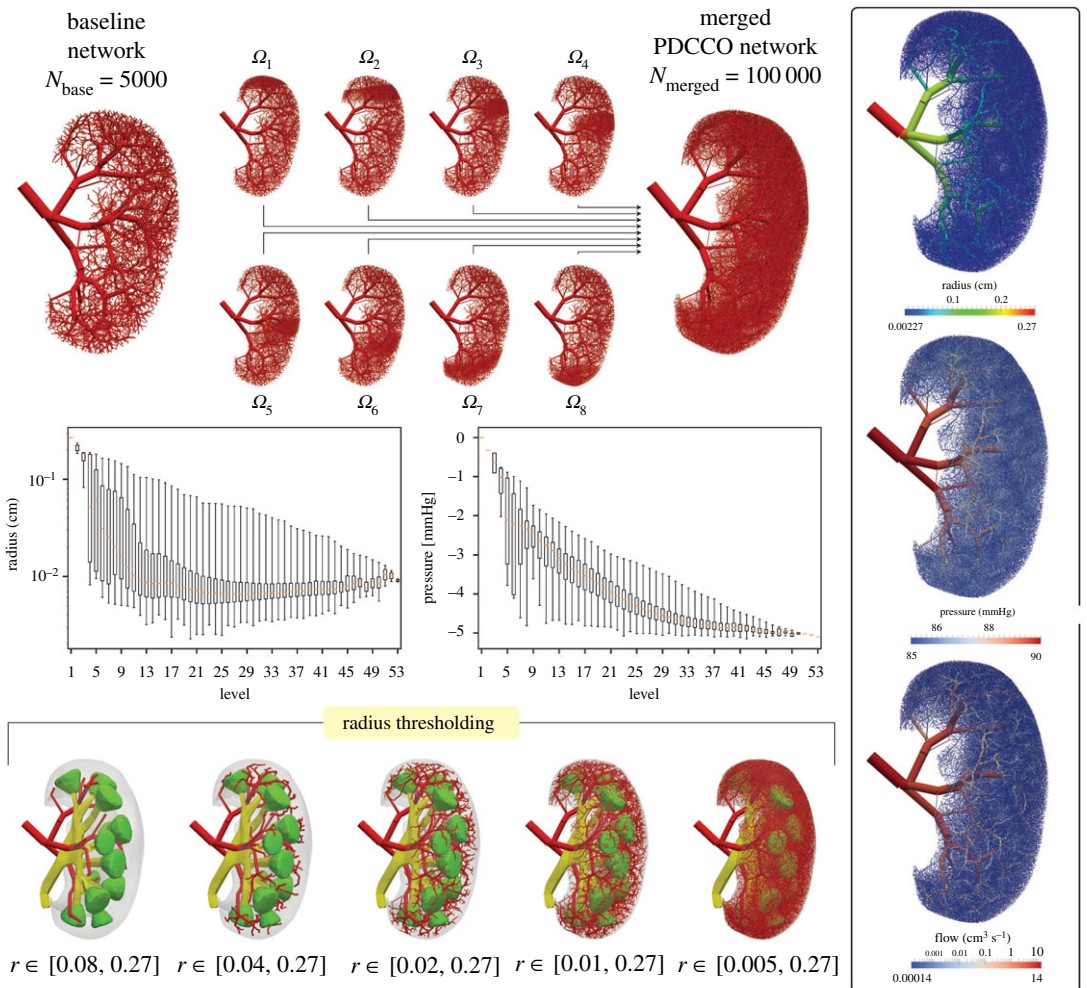

**Figure 11.** Vascularization of a human kidney model using the PDCCO algorithm. The vascularization is initiated with a baseline network and a partition into eight subdomains (top panel). The total number of terminal vessels reaches 100 000. Radius and pressure profiles in the network are reported as a function of the vessel generation (middle panel). Also, the spatial features of the vessel radius, blood pressure and flow rate are reported (right panel). From the merged network, thresholding using the radius variable is applied to illustrate the different scales in the model (bottom panel).

planes along the vertical coordinate. For each of these subdomains, the number of terminal segments added were $N_1 = 7742$, $N_2 = 13\,870$, $N_3 = 11\,809$, $N_4 = 11\,942$, $N_5 = 13\,803$, $N_6 = 12\,958$, $N_7 = 14\,668$, $N_8 = 8208$. The flow rate and radius at the inlet of the network were set to $Q_{in} = 14\ \mathrm{cm}^3\,\mathrm{s}^{-1}$ and $r_1 = 0.27$ cm.

Figure 11 illustrates the result of the PDCCO algorithm for the vascularization of a human kidney. On the top of the figure, the baseline network containing 5000 vessels is shown. This network was generated sequentially. After this first stage, the vascularization is parallelized into eight subdomains. The individual vascular networks are also shown next to the baseline network. Finally, the merging procedure is illustrated on the right, where the final network is assembled, reaching 100 000 terminal vessels. In the middle of the same figure, the radius and pressure profiles are displayed as a function of vessel generation. The pressure drop in the network is around 5 mmHg. On the right, the radius, pressure, and flow rate are displayed across the entire tree. Finally, at the bottom of the figure, we report a sequence of panels that allows us to visualize the resulting network structure by thresholding the merged tree with the vessel radius. Note the hierarchical arborization surrounding the pelvis and pyramids to finally reach the cortex, where the arterial vessels are closely packed.

In figure 12, we show several quantities classified in terms of the Strahler order. Specifically, that figure displays the total cross-sectional lumen area associated with all the vessels with a given Strahler order, the degree of connectivity of vessels, the pressure and flow rate as a function of the Strahler order, and, finally, the intravascular volume of the subtended tree to a vessel of a certain radius.

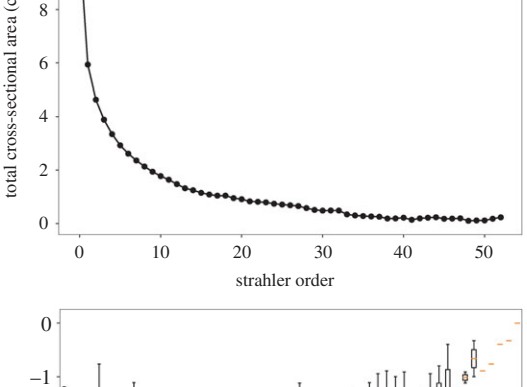
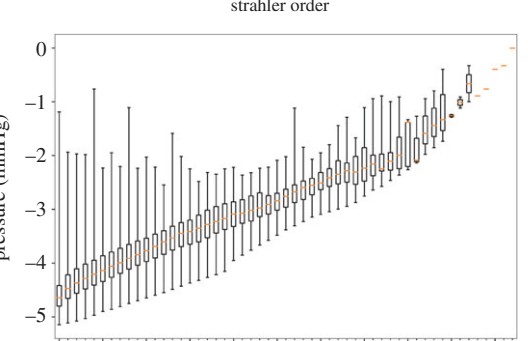
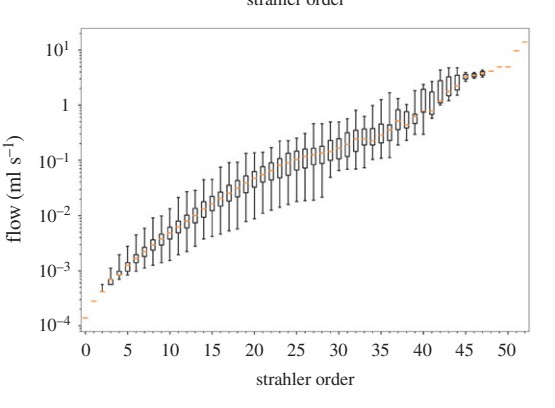
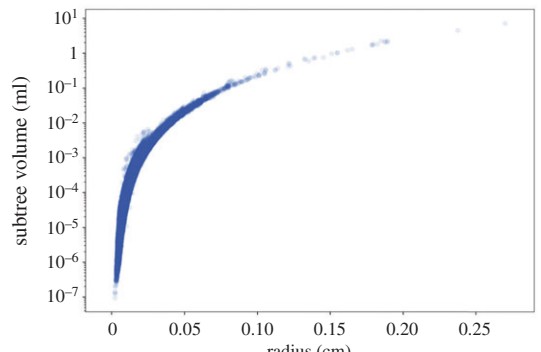

**Figure 12.** Geometric and functional characterization of the renal vascular network in terms of the Strahler order (root vessel connected to vessel with order −1). In lexicographic order, we report: combined cross-sectional lumen area, network connectivity, pressure and flow rate as a function of the Strahler order. In the bottom row, we report the subtended tree volume as a function of the vessel radius.

# 4. Discussion

The relevance of generating networks of vessels to vascularize entire organs or even portions of tissues has turned into a valuable tool for research in drug delivery [42], and functional assessment of liver [43], heart [44], brain [23,45,46], among other applications. Thus, the ability of automatic algorithms to cover vascular territories with ensembles of vessels started to be tested in increasingly realistic applications and challenging problems in the field of biomedical engineering. CCO strategies emerged as a promising approach to address this problem [24], leading to variants, such as the DCCO approach [30] to enhance the capabilities of the methodology to deal with the construction of intricate vascular networks. The intrinsic limitation of the sequential CCO-based approaches is the limited number of vessels that the algorithm can generate in a reasonable timeframe. In this work, the main contribution was the proposition of a parallel algorithm consisting of three steps: (i) the generation of a baseline network using the DCCO approach; (ii) the partitioning of the vascular domain and the parallel application of the DCCO method to separately grow these networks and (iii) the merging step in which all networks are joined, and the complete network is adequately scaled to comply with the geometric constraints and the branching power law.

In this work, we first proposed experiments to demonstrate that the PDCCO networks were in close agreement with DCCO networks in terms of geometry and function. We have seen that the vessel radius and length were in full agreement for both parallel and sequential algorithms. That was demonstrated for different numbers of segments in the baseline network and different subdomain shapes. These experiments were relevant to understanding the limitations in the setting of the parallel approach. Hence, we saw that even with baseline networks featuring a small number of segments, the error in the intravascular volume in PDCCO networks remained below 7% on average, while the radius and vessel length featured a similar distribution to DCCO networks. Regarding the aspect ratio of the partitions, the differences were also small.

Despite such a difference in vessel radius as a function of the depth in the vascular network (vessel generations), the computation of the pressure profile from proximal to distal locations does not change significantly between PDCCO and DCCO generated networks. The differences observed in the pressure drop were always below 1 mmHg on average. Pressure drops faster for PDCCO for deeper domains (i.e. $A_r > 1$) but it is quite consistent for broad and cubic-like domains (i.e. $A_r \leq 1$). Thus, the partitioning of the domain should favour the creation of subdomains of regular dimensions or transversally larger sections (concerning the perfusion inlets) to avoid a bias of the pressure drop due to the parallelization.

At this point, we should emphasize that the embarrasingly parallel nature of the proposed algorithm does not support global optimality in the sense of conventional CCO-based algorithms. As a matter of fact, the original CCO approach does not yield a true optimal solution either. Indeed, in the sequential approach, the current state of the network is fixed, and only local bifurcation perturbation and whole-network diameter scaling are considered. In [47], the authors discuss what they call post-optimization, in which the state of the network, after connecting a new segment, is modified by altering all the bifurcation points. This enables the algorithm to reach a better degree of optimality. However, as noted by these authors, even if a more optimal network is achieved, the functional significance of these modifications (pressure profiles) remains unaffected. The optimality of a certain network depends mainly on the placement of the first vessel generations [47]. Moreover, post-optimization adds significant computational burden, and is not practical for large problems. As in the original CCO method, in our approach, the optimality is mostly dominated through the first steps of the algorithm, when generating the baseline tree, which is accomplished through the conventional sequential algorithm (see §3.1).

The proposed PDCCO strategy can also be understood as a multiscale method for the generation of vascular networks. In fact, the baseline tree accounts for the macro-scale features of the vasculature, while the remainder of the network architecture responds to local micro-scale phenomena. In turn, the point at which the partitioning procedure is executed, after the baseline tree is generated, can be seen as the point at which we assume that the scale separation hypothesis holds. The multiscale nature of the approach is also evidenced in the experiments reported in §3.3 where the baseline tree topology was determined by the configuration of the cost functional, while the partitioned algorithm dealt with the micro-scale vascular features in a parallel fashion regardless of the macro-scale connectivity. Although there is no definite answer about which is the stage at which scale separation can be safely assumed, numerical experiments reported in this study evidenced a negligible departure from the optimal solution delivered by the sequential algorithm, both in terms of geometry and function. In turn, the merging procedure can be considered as a consistency step, in which the entire tree geometrically accommodates to the small-scale perturbations introduced in the micro-scale vascularization processes.

Concerning the scalability of the PDCCO method, specifically, the ability to efficiently generate extensive vascular networks, we have proposed a test to vascularize a domain with 100 000 vessels, using partitions with a different number of subdomains. In such a test, the construction of the baseline network is common to all the experiments, and the real potential of the parallel approach is seen in the phase where the baseline network is grown to the full merged network. The test allowed us to exploit the embarrasingly parallel nature of the proposed algorithm to reach a vascular network of 100 000 terminal vessels (that is 200 000 segments in the network). Removing the cost in the baseline network generation, the time spent in building the network scaled linearly with the number of partitions, and the geometric (intravascular volume and vessel radius profile) and functional (blood pressure profile) features of the resulting networks remained on an equal footing.

Finally, we have employed the proposed algorithm to build the entire vascular network in a prototypical human kidney. While previous efforts in this enterprise were reported in the literature [48], this has been the first attempt to build the entire renal vasculature from scratch to the level of interlobular arteries. Only by using the parallel approach was it possible to reach the level of

interlobular arteries. Although the morphometric validation of the so-constructed vascular model is not the focus of the present paper, it is possible to note the physiological morphometric (vessel length and radius) and functional (pressure and flow rate) features in the model. In fact, note that it was not until only recently that reliable three-dimensional data about the rat renal vasculature was reported [49]. By using a multi-stage approach, the algorithmic generation was initiated with three segmental vessels, to vascularize the renal pelvis region in the first stage. The subsequent vascularization of the cortical tissue was developed until reaching a baseline network with 5000 terminal segments. This was done using the sequential DCCO algorithm. Then, the domain was partitioned, and the PDCCO algorithm was applied until reaching the goal of 100 000 terminal vessels (i.e. 200 000 segments overall). In this process, the first generations of segmental vessels were automatically placed in the renal core, and from there, the interlobar arteries, the arcuate vessels, and finally, the interlobular arteries. The perfusion pressure in the vascular segments generated with the proposed approach are in accordance with the measurements of intra-renal pressures assessed *in vivo* [50]. Even with an arborization involving tens of thousands of vascular ramifications from the renal artery, the perfusion pressure slightly changes. Substantial pressure drops occur only in the afferent arteriole segment, immediately before the entry into the glomerulus [50]. The construction of such a complex architectural arrangement of vessels was only possible due to the combination of the parallel and multi-stage approaches, paving the way towards an integrative analysis of the macrocirculation, microcirculation and cellular physiology for the description of whole-organ kidney function.

In [51], the authors characterized the morphology of the rat renal vasculature using micro-computer tomography images. They provided a comprehensive analysis of both arterial and venous networks, reporting different geometric network characteristics as a function of the Strahler order. In the present study, for the prototypical model of the kidney vasculature created using the proposed PDCCO, we also classified the geometric and functional entities associated with the model in terms of the Strahler order. The network behaviour yields an asymptotic growth of the cross-sectional lumen area as the Stralher order approaches zero. Flow rate drops more than linearly (log-scale), and the connectivity matrix reveals a coupling between small vessels and high Strahler order vessels. As a consequence, even if the pressure drops almost linearly with the Stralher order, there are small vessels which are exposed to high blood pressure. This may have an important role in the damage suffered by glomeruli (connected right distal to these smaller vessels) in hypertensive scenarios, even if the flow rate is small. In turn, the relation between the subtended intravascular volume tree and the vessel radius is consistent with the data reported in [51] (cf. Figure 11).

In terms of model validation, there is scarcity of data in the field. One remarkable exception is [51], which studied the vasculature in the rat kidney. This speaks about the highly challenging enterprise that carrying such a study implies. In view of the lack of access to anatomical data, we focused on the description and validation of an efficient technique that enables the generation of massive networks. To do that, we rely on the already demonstrated capabilities of one of the most well-established approaches available in the literature, the CCO approach. Based on the CCO methodology, whose physiological and anatomical consistency was already demonstrated in numerous studies [24,25,52–56], we proposed an algorithmic modification that allowed us to scale the number of vascular segments. Here, we proposed a strategy to remove an algorithmic barrier from CCO, and communicated evidence in favour of an efficient technique, which can perform equivalently to a standard and well-established one.

As a limitation of the proposed strategy, we can mention that the algorithm may suffer from a lack of convergence (rare but possible) when merging the networks in the case of nonlinear rheological blood behaviour. Another limitation we can mention is the degeneration of vascular networks in extremely degenerated subdomains. The latter can be mitigated by performing regular partitions of the domain to be vascularized.

# 5. Final remarks

This work presented a novel strategy to exploit the parallel generation of extensive vascular networks by using an existing DCCO algorithm. The parallel approach removed limitations inherent to the sequential algorithm, enabling the automatic generation of vascular networks containing hundreds of thousands of vascular segments.

One of the relevant contributions of the present work is the ability to create complex networks that integrate the macro- and micro-circulation realms, thus bridging the different scales featured by the CVS.

Furthermore, we reported the *in silico* vascularization of a prototypical geometry of the human kidney for the first time. This result was achieved by integrating a multi-stage strategy and the parallel generation of vascular networks, starting at the renal artery and reaching the scale of interlobular vessels.

Data accessibility. Data and relevant code for this research work are stored in GitHub: https://github.com/lfmc/pdccoRSOS and have been archived within the Zenodo repository: https://doi.org/10.5281/zenodo.5529331 Execution scripts, input files and generated output files are available within Dryad: https://doi.org/10.5061/dryad.t4b8gtj25.

Authors' contributions. L.F.M.C. contributed by implementing and conceptualizing PDCCO, writing a first draft of the manuscript, generating the results and participating in their analysis. G.D.M.T. contributed in the conceptualization, research design, formal analysis of the results and participated in the discussions of the manuscript. G.D.M.T. implemented the DCCO library (VItA library) used in this work. M.Y.-I contributed by interpreting the data and analysing the implications. P.J.B. contributed in the conceptualization of the PDCCO, research design, results interpretation and leading the manuscript writing. All authors (L.F.M.C., G.D.M.T., M.Y.-I. and P.J.B.) provided critical feedback and helped shape the research, analysis and manuscript.

Competing interests. We declare we have no competing interests.

Funding. This project was partially supported by the Brazilian agencies CNPq (grant no. 407751/2018-1), and FAPESP (grant no. 2014/50889-7). G.D.M.T. acknowledges the support of the Li Ka Shing foundation via the philanthropic Li Ka Shing grant no. 9077/31/8402.

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
