## [Peer Review File · Royal Society Open Science]

Review History

RSOS-210973.R0 (Original submission)

Review form: Reviewer 1

Is the manuscript scientifically sound in its present form?

No

Are the interpretations and conclusions justified by the results?

Yes

Is the language acceptable?

Yes

Do you have any ethical concerns with this paper?

No

Have you any concerns about statistical analyses in this paper?

No

Recommendation?

Major revision is needed (please make suggestions in comments)

Comments to the Author(s)

This paper needs revisions in order to increase its possible interest toward readers that are not necessarily expert in CCO. I find the comparison of the sequential and parallel version of the CCO approach too detailed and I suggest instead to explore better the anatomical coherence of the constructed model

See the attached file for details.

Please see attached file (Appendix A).

Review form: Reviewer 2

Is the manuscript scientifically sound in its present form?

No

Are the interpretations and conclusions justified by the results?

Yes

Is the language acceptable?

Yes

Do you have any ethical concerns with this paper?

No

Have you any concerns about statistical analyses in this paper?

No

Recommendation?

Major revision is needed (please make suggestions in comments)

Comments to the Author(s)

This study proposed a parallel computing algorithm to develop a vascular network based on constrained constructive optimization (CCO) algorithm. One drawback of the CCO algorithm is that the vascular network has to be sequentially updated in the entire computational domain, and thus CCO algorithm requires much computational cost in large network creation. The authors proposed a simple solution: the computational domain was decomposed in multiple local domains, allowing the network updates as an embarrassingly parallel task. The authors demonstrated practical consistencies between the original sequential algorithm and the proposed parallel algorithm through several numerical cases; however, such a simple algorithm requires additional artificial processing and raises severe concerns to violate the rationale of the original CCO algorithm from viewpoints of mathematical optimization. Before publication, the authors should carefully discuss these issues and clarify specific usages of their proposed algorithm following the above limitations.

Comment 1:

In my understanding, numerical algorithms of the vascular network creation cannot guarantee the perfect consistencies of the anatomical characteristics due to its complexities, including unknown factors. From this viewpoint, the original CCO algorithm was proposed to provide a vascular network as a result of the global optimal solution of some cost functions and constraint

conditions based on physiological knowledge. The sequential updating scheme, which is one of the drawbacks of the CCO algorithm, is thus required to satisfy global optimalities. In contrast, the proposed parallel algorithm does not support global optimalities and independently satisfies local optimalities in each artificially decomposed domain independently. These theoretical inconsistencies require other rationales of the proposed parallel algorithm and created vascular networks.

Comment 2:

As the authors stated in final remarks that “One of the relevant contributions of the present work is the ability to create complex networks able to integrate the macro and micro-circulation realms”, I suppose this parallel algorithm aims to simplify the micro-scale network creation process, which is generated around base tree structure and not affected the global network structure. If so, this strategy should be explicitly stated as the aim of this proposed algorithm, in the introduction and methods sections (and hopefully also in the abstract).

Moreover, this parallel algorithm implicitly considers multiscale properties of the network, and thus has the lower limit about the consistencies of the base tree geometry, dependent on the size of entire and decomposed computational domains. Further evaluations by extending numerical experiments (a) and (b) regarding this issue may clarify both the effectiveness and limitations of this parallel algorithm. In other words, although numerical experiments (a) and (b) provided consistent properties between original and proposed algorithms from limited case studies, I hope the authors clarify the limitations of the size of the baseline tree and decomposition.

Comment 3:

In the section of the experiment (d), more description about vascular network creation is needed. I suppose that the authors applied a staged algorithm to create the vascular network consistent with anatomical knowledge. How long is the radius of the baseline network? How did the authors decompose the computational region? In discussion, the authors described that “In this process, the first generations of segmental vessels were automatically placed in the renal core, and from there, the interlobar arteries, the arcuate vessels, and finally, the interlobular arteries,” however, the section of the experiment (d) does not explicitly show how the authors achieved this process. I suppose that these multiscale vasculatures seem to be generated mainly by the staged algorithm, and the proposed parallel algorithm was exclusively used in microscale vasculatures formation. The main contribution of the proposed parallel algorithm in this multiscale vascular network formation should be clarified.

Comment 4:

The size of the figures is too small to read this paper in print. Even on the monitor, I cannot read the words in figures without magnifying the pdf to at least 400%.

Comment 5:

Figure 4 in line 33 on page 5 and line 8 on page 6 may be Figure 1.

Decision letter (RSOS-210973.R0)

Dear Dr Maso Talou,

The Editors assigned to your paper RSOS-210973 "Parallel generation of extensive vascular networks with application to an archetypal human kidney model" have now received comments from reviewers and would like you to revise the paper in accordance with the reviewer comments and any comments from the Editors. Please note this decision does not guarantee eventual acceptance.

Please submit your revised manuscript and required files (see below) no later than 21 days from today's (ie 16-Aug-2021) date. Note: the ScholarOne system will 'lock' if submission of the revision is attempted 21 or more days after the deadline. If you do not think you will be able to meet this deadline please contact the editorial office immediately.

on behalf of Dr Peter Stewart (Associate Editor) and R. Kerry Rowe (Subject Editor)
openscience@royalsociety.org

Reviewer comments to Author:

Reviewer: 1

Comments to the Author(s)

This paper needs revisions in order to increase its possible interest toward readers that are not necessarily expert in CCO. I find the comparison of the sequential and parallel version of the CCO approach too detailed and I suggest instead to explore better the anatomical coherence of the constructed model

See the attached file for details (Review_Cury_et_al.pdf)

Reviewer: 2

Comments to the Author(s)

This study proposed a parallel computing algorithm to develop a vascular network based on constrained constructive optimization (CCO) algorithm. One drawback of the CCO algorithm is

that the vascular network has to be sequentially updated in the entire computational domain, and thus CCO algorithm requires much computational cost in large network creation. The authors proposed a simple solution: the computational domain was decomposed in multiple local domains, allowing the network updates as an embarrassingly parallel task. The authors demonstrated practical consistencies between the original sequential algorithm and the proposed parallel algorithm through several numerical cases; however, such a simple algorithm requires additional artificial processing and raises severe concerns to violate the rationale of the original CCO algorithm from viewpoints of mathematical optimization. Before publication, the authors should carefully discuss these issues and clarify specific usages of their proposed algorithm following the above limitations.

Comment 1:

In my understanding, numerical algorithms of the vascular network creation cannot guarantee the perfect consistencies of the anatomical characteristics due to its complexities, including unknown factors. From this viewpoint, the original CCO algorithm was proposed to provide a vascular network as a result of the global optimal solution of some cost functions and constraint conditions based on physiological knowledge. The sequential updating scheme, which is one of the drawbacks of the CCO algorithm, is thus required to satisfy global optimalities. In contrast, the proposed parallel algorithm does not support global optimalities and independently satisfies local optimalities in each artificially decomposed domain independently. These theoretical inconsistencies require other rationales of the proposed parallel algorithm and created vascular networks.

Comment 2:

As the authors stated in final remarks that "One of the relevant contributions of the present work is the ability to create complex networks able to integrate the macro and micro-circulation realms", I suppose this parallel algorithm aims to simplify the micro-scale network creation process, which is generated around base tree structure and not affected the global network structure. If so, this strategy should be explicitly stated as the aim of this proposed algorithm, in the introduction and methods sections (and hopefully also in the abstract). Moreover, this parallel algorithm implicitly considers multiscale properties of the network, and thus has the lower limit about the consistencies of the base tree geometry, dependent on the size of entire and decomposed computational domains. Further evaluations by extending numerical experiments (a) and (b) regarding this issue may clarify both the effectiveness and limitations of this parallel algorithm. In other words, although numerical experiments (a) and (b) provided consistent properties between original and proposed algorithms from limited case studies, I hope the authors clarify the limitations of the size of the baseline tree and decomposition.

Comment 3:

In the section of the experiment (d), more description about vascular network creation is needed. I suppose that the authors applied a staged algorithm to create the vascular network consistent with anatomical knowledge. How long is the radius of the baseline network? How did the authors decompose the computational region? In discussion, the authors described that "In this process, the first generations of segmental vessels were automatically placed in the renal core, and from there, the interlobar arteries, the arcuate vessels, and finally, the interlobular arteries," however, the section of the experiment (d) does not explicitly show how the authors achieved this process. I suppose that these multiscale vasculatures seem to be generated mainly by the staged algorithm, and the proposed parallel algorithm was exclusively used in microscale vasculatures formation. The main contribution of the proposed parallel algorithm in this multiscale vascular network formation should be clarified.

Comment 4:

The size of the figures is too small to read this paper in print. Even on the monitor, I cannot read the words in figures without magnifying the pdf to at least 400%.

Comment 5:

Figure 4 in line 33 on page 5 and line 8 on page 6 may be Figure 1.

===PREPARING YOUR MANUSCRIPT===

===PREPARING YOUR REVISION IN SCHOLARONE===

Author's Response to Decision Letter for (RSOS-210973.R0)

See Appendices B & C.

RSOS-210973.R1 (Revision)

Review form: Reviewer 1

Is the manuscript scientifically sound in its present form?

Yes

Are the interpretations and conclusions justified by the results?

Yes

Is the language acceptable?

Yes

Do you have any ethical concerns with this paper?

No

Have you any concerns about statistical analyses in this paper?

No

Recommendation?

Accept as is

Comments to the Author(s)

The version I was sent to revise is not compiled correctly, since all the bibliogr references were question marks.

Review form: Reviewer 2

Is the manuscript scientifically sound in its present form?

Yes

Are the interpretations and conclusions justified by the results?

Yes

Is the language acceptable?

Yes

Do you have any ethical concerns with this paper?

No

Have you any concerns about statistical analyses in this paper?

No

Recommendation?

Accept as is

Comments to the Author(s)

Thank you for addressing my concerns and clarifying the concept of parallelized modeling of the vascular network. I support publishing the paper in its current form.

Decision letter (RSOS-210973.R1)

Dear Dr Maso Talou,

It is a pleasure to accept your manuscript entitled "Parallel generation of extensive vascular networks with application to an archetypal human kidney model" in its current form for publication in Royal Society Open Science. The comments of the reviewer(s) who reviewed your manuscript are included at the foot of this letter.

The proof of your paper will be available for review using the Royal Society online proofing system and you will receive details of how to access this in the near future from our production office (opencscience_proofs@royalsociety.org). We aim to maintain rapid times to publication after acceptance of your manuscript and we would ask you to please contact both the production office and editorial office if you are likely to be away from e-mail contact to minimise delays to publication. If you are going to be away, please nominate a co-author (if available) to manage the proofing process, and ensure they are copied into your email to the journal.

Kind regards,
Royal Society Open Science Editorial Office
Royal Society Open Science
opencscience@royalsociety.org

on behalf of Dr Peter Stewart (Associate Editor) and R. Kerry Rowe (Subject Editor)
opencscience@royalsociety.org

Associate Editor Comments to Author (Dr Peter Stewart):

Comments to the Author:

Dear authors, thank you for revising your manuscript. Both referees were satisfied with your updates and responses, so I am happy to accept your manuscript as it stands.

Reviewer comments to Author:

Reviewer: 1

Comments to the Author(s)

The version I was sent to revise is not compiled correctly, since all the bibliogr references were question marks.

Reviewer: 2

Comments to the Author(s)

Thank you for addressing my concerns and clarifying the concept of parallelized modeling of the vascular network. I support publishing the paper in its current form.

Appendix A

Reviewer's report on the paper

“Parallel generation of extensive vascular networks with application to an archetypal human kidney model”

by L.F.M. Cury, G.D. Maso Talou, M. Younes-Ibrahim and P.J. Blanco

Aim of this work is to propose a scalable parallel version of the CCO algorithm, called PDCCO, to generate in silico microvascular trees. The authors prove that the present approach allows to generate in a modest amount of time vascular trees which show limited differences from the trees generated with the corresponding sequential approach. Moreover, as a realistic study case, they build a complete vascularization tree for a prototypal human kidney.

My opinion is that this paper is an interesting work, well explained and clearly illustrated. Its main limitation lays in the fact that the work is too “self-referential” all inside the context of CCO methods: the CCO approach and related modifications may be for sure promising techniques to obtain in silico microvascular trees, but no paper on this subject can avoid a morphometric validation of the constructed vascular models accompanied by a physiological study in comparison to in vivo (or in vitro if available) measurements. This allows to assess the anatomical coherence of the artifact as well as the plausibility of the computed fields as representative of real fluid-dynamical fields (even if the authors do offer some comments for this latter point).

For these reason, I believe that revisions are required so that less emphasis should be paid to the internal details of the parallel algorithm (possibly moving them to an Appendix) and opening more space to a validation against realistic data.

Here below a more detailed discussion of the critical points:

- in the Introduction, a bibliographical review of the state of the art of in silico microvessel generation is performed. However, the choice of the cited papers is rather biased and it does not include contributions that are relevant to this topic, for example those using Delaunay tessellations by the group of Lorthois and colleagues
- what about the number of branches that sprout from a parent vessels? Are there mainly bifurcations? Is this parameter controllable?
- Table 2 reports a comparison with some realistic morphological data, but the chosen parameters seem to offer a limited analysis of the generated model, only based on a comparison of volumetric data and not investigating volume/to surface ratios, radii distribution, mean distance to tissue
- how does the introduction of non-linear models of blood rheology impacts the convergence of the PCCO algorithm? This aspect is quickly addressed at the very end of the paper but it is rather important since at this vessel size the peculiar features of blood rheology significantly impact the overall flow
- how the choice of the cost function impacts the generated tree and - possibly – the stability of the algorithm?
- how can the algorithm handle multiple input/output vascular trees?

Minor comments:

- line 15: this sentence does not seem to be formed with an appropriate verb
- line 39: stand → stands

- more details could be useful about the approach used to obtain branching in the renal vascular tree

Appendix B

Response to Reviewer #1

Parallel generation of extensive vascular networks with application to an archetypal human kidney model

Luis F.M. Cury, Gonzalo D. Maso Talou, Mauricio Younes-Ibrahim and Pablo J. Blanco

Authors' general comment:

The reviewer will find attached a revised version of the manuscript addressing the concerns, comments and suggestions made by the reviewers. In **blue** and **red** we highlight modifications attending comments from Reviewer #1 and #2, respectively. The modifications are detailed in the specific rejoinder to each reviewer. Finally, we would like to acknowledge the positive contribution of the reviewers' comments to the manuscript's quality.

Reviewer's general comment.

My opinion is that this paper is an interesting work, well explained and clearly illustrated. Its main limitation lays in the fact that the work is too "self-referential" all inside the context of CCO methods: the CCO approach and related modifications may be for sure promising techniques to obtain *in silico* microvascular trees, but no paper on this subject can avoid a morphometric validation of the constructed vascular models accompanied by a physiological study in comparison to *in vivo* (or *in vitro* if available) measurements. This allows to assess the anatomical coherence of the artifact as well as the plausibility of the computed fields as representative of real fluid-dynamical fields (even if the authors do offer some comments for this latter point).

For these reason, I believe that revisions are required so that less emphasis should be paid to the internal details of the parallel algorithm (possibly moving them to an Appendix) and opening more space to a validation against realistic data.

Response to the reviewer's global appraisal:

We thank the reviewer for the in-depth appraisal of our work. As the reviewer pointed out, comparison with physiological measurements is critical to gain insight into the model's capabilities and limitations. Nonetheless, the paucity of data in the field is

remarkable. Up to our knowledge, only a few groups have been able to carry out this kind of comparison with corrosion casts and with data extracted from animal models. Schreiner's group, Kassab's group, Lorthois' group, and Linninger's group are the ones that we are aware of. The scarcity of data and scientific publications speaks about the highly challenging enterprise that carrying such a study implies. We have tried our best to collect data, but we had no access to resources to accomplish that task. This is the reason for which we focused on the description and validation of a technique that enables the generation of massive networks.

To do that, we rely on the already demonstrated capabilities of one of the well-established approaches available in the literature, the CCO approach. Based on the CCO methodology, whose physiological and anatomical consistency was already demonstrated, we proposed an algorithmic modification that allows us to scale the number of segments. Indeed, the proposed methodology is ahead of the type of data that allows a refined model validation. Notwithstanding this, we genuinely believe that the advancement of the field should also be constituted of technical papers that remove algorithmic barriers and communicate evidence in favour of efficient novel techniques, which, at the same time, can perform equivalently to more standard and well-established ones. We have incorporated in the Discussion section a paragraph on this aspect, to clarify the reader the scope and limitations of the proposed approach.

In the revised version of the manuscript, we have addressed the specific concerns listed below. Moreover, we have improved the reporting of data for the kidney section. We have followed ideas according to Nordsletten et al. (Structural morphology of renal vasculature), and now we report the data similarly to the morphometric analysis of the vasculature in the mice kidney performed in that paper.

Specific response to the reviewer's comments:

1. in the Introduction, a bibliographical review of the state of the art of in silico microvessel generation is performed. However, the choice of the cited papers is rather biased and it does not include contributions that are relevant to this topic, for example those using Delaunay tessellations by the group of Lorthois and colleagues

Response: We have expanded the bibliographic review in the introduction as pointed out by the reviewer.

2. what about the number of branches that sprout from a parent vessels? Are there mainly bifurcations? Is this parameter controllable?

Response: In the proposed algorithm, we work with binary trees, as explained in the first line of Section 2(a). Although this seems to be a limitation, it is a reasonable hypothesis at least for networks of small arteries and arterioles.

3. Table 2 reports a comparison with some realistic morphological data, but the chosen parameters seem to offer a limited analysis of the generated model, only based on a comparison of volumetric data and not investigating volume/to surface ratios, radii distribution, mean distance to tissue

Response: The data reported in Table 2 is a description of the anatomical domain used as input data to the algorithm. For the sake of clarity, we have rewritten the caption for that table, and also included an expanded description at the end of the first paragraph in Section 3(e) in the new version of the manuscript.

4. how does the introduction of non-linear models of blood rheology impacts the convergence of the PCCO algorithm? This aspect is quickly addressed at the very end of the paper but it is rather important since at this vessel size the peculiar features of blood rheology significantly impact the overall flow

Response: We agree with the reviewer that proper modelling of the complex blood rheology at this scale is of utmost importance, as plasma skimming and Fahraeus-lindqvist effect have a clear impact in blood viscosity. In the new version of the manuscript, we have incorporated a more detailed comment on the choice of the viscosity model, as can be seen in the last part of Section 2(a). There, we clarify that for the experiments performed the fixed-point approach used to deal with nonlinearities is stable and convergent. Also, we comment that this approach may not be convergent for unrealistically small vessels (smaller than capillaries).

5. how the choice of the cost function impacts the generated tree and - possibly - the stability of the algorithm?

Response: We have addressed this issue by incorporating in the revised manuscript a section with numerical experiments that show the impact of the cost

functional in the resulting vascular network (see the new Section 3(c)). Specifically, we propose to vascularise a domain with multiple inlets, and investigate the lack of sub-tree balance, in terms of flow rate carried by each inlet, as a function of the parameters that define the cost functional.

6. how can the algorithm handle multiple input/output vascular trees?

Response: The proposed approach can handle vascular domains featuring multiple inlets. In the revised manuscript we have included an additional example (see previous comment, and the new Section 3(c)) which serves to analyse the impact of the definition of the cost functional in the balancing of the different downstream networks associated to each one of the inlets.

Specific response to the reviewer's minor comments

1. line 15: this sentence does not seem to be formed with an appropriate verb

Response: The sentence was rewritten to express more clearly the concept.

2. line 39: stand \rightarrow stands

Response: The sentence was rewritten to accommodate the revised bibliographic review.

3. more details could be useful about the approach used to obtain branching in the renal vascular tree

Response: We have expanded the description of the stages used to construct the vascularisation of the kidney model, as seen in Section 3(e) in the new version of the manuscript.

We thank the reviewers for their contribution to our manuscript.
With best regards,

The authors

Appendix C

Response to Reviewer #2

Parallel generation of extensive vascular networks with application to an archetypal human kidney model

Luis F.M. Cury, Gonzalo D. Maso Talou, Mauricio Younes-Ibrahim and Pablo J. Blanco

Authors' general comment:

The reviewer will find attached a revised version of the manuscript addressing the concerns, comments and suggestions made by the reviewers. In **blue** and **red** we highlight modifications attending comments from Reviewer #1 and #2, respectively. The modifications are detailed in the specific rejoinder to each reviewer. Finally, we would like to acknowledge the positive contribution of the reviewers' comments to the manuscript's quality.

Reviewer's general comment.

This study proposed a parallel computing algorithm to develop a vascular network based on constrained constructive optimization (CCO) algorithm. One drawback of the CCO algorithm is that the vascular network has to be sequentially updated in the entire computational domain, and thus CCO algorithm requires much computational cost in large network creation. The authors proposed a simple solution: the computational domain was decomposed in multiple local domains, allowing the network updates as an embarrassingly parallel task. The authors demonstrated practical consistencies between the original sequential algorithm and the proposed parallel algorithm through several numerical cases; however, such a simple algorithm requires additional artificial processing and raises severe concerns to violate the rationale of the original CCO algorithm from viewpoints of mathematical optimization. Before publication, the authors should carefully discuss these issues and clarify specific usages of their proposed algorithm following the above limitations.

Response to the reviewer's global appraisal:

We thank the reviewer for the in-depth appraisal of our work. In the revised version of the manuscript, we have addressed your specific concerns listed below.

Specific response to the reviewer’s major comments

1. In my understanding, numerical algorithms of the vascular network creation cannot guarantee the perfect consistencies of the anatomical characteristics due to its complexities, including unknown factors. From this viewpoint, the original CCO algorithm was proposed to provide a vascular network as a result of the global optimal solution of some cost functions and constraint conditions based on physiological knowledge. The sequential updating scheme, which is one of the drawbacks of the CCO algorithm, is thus required to satisfy global optimalities. In contrast, the proposed parallel algorithm does not support global optimalities and independently satisfies local optimalities in each artificially decomposed domain independently. These theoretical inconsistencies require other rationales of the proposed parallel algorithm and created vascular networks.

Response: We agree with the reviewer that the proposed algorithm does not support global optimality. We have expanded the discussion section in the new version of the manuscript to address this topic. Indeed, the original CCO approach does not either. Indeed, in the sequential approach, the current state of the network is fixed, and only local bifurcation perturbation and whole-network diameter scaling are considered. Furthermore, since the optimisation problem is non-smooth as a result of the topological changes introduced at each generation step, a sequential algorithm cannot reach a genuinely optimal solution (not even the true biological process is an optimisation problem, as acknowledged by the reviewer when referring to “unknown factors”). What it has been demonstrated, is that CCO-based approaches provide anatomically and physiologically realistic vascular networks. In one of the early CCO studies by Karch et al. (“Three-Dimensional Optimisation of Arterial Tree Model”), the authors discuss what they call post-optimisation, in which the state of the network, after connecting a new segment, is modified by altering all the bifurcation points. This enables the algorithm to reach a better degree of optimality. However, as noted by these authors, even if a more optimal network is achieved, the functional significance of these modifications remains unaffected. Moreover, the optimality depends mainly on the first stages in the network generation, as reported by the same authors. Post-optimisation was also regarded as a computationally expensive strategy, not applicable to large problems. Hence, as in the original CCO method, the optimality in our approach is mostly dominated through the first steps of the algorithm when generating the baseline tree, which is accomplished through the

conventional sequential algorithm. This has been the purpose of the results reported in Section 3(a), where the size of the baseline tree has been taken as a parameter from where to launch the partitioned vascularisation.

2. As the authors stated in final remarks that “One of the relevant contributions of the present work is the ability to create complex networks able to integrate the macro and micro-circulation realms”, I suppose this parallel algorithm aims to simplify the micro-scale network creation process, which is generated around base tree structure and not affected the global network structure. If so, this strategy should be explicitly stated as the aim of this proposed algorithm, in the introduction and methods sections. Moreover, this parallel algorithm implicitly considers multiscale properties of the network, and thus has the lower limit about the consistencies of the base tree geometry, dependent on the size of entire and decomposed computational domains. Further evaluations by extending numerical experiments (a) and (b) regarding this issue may clarify both the effectiveness and limitations of this parallel algorithm. In other words, although numerical experiments (a) and (b) provided consistent properties between original and proposed algorithms from limited case studies, I hope the authors clarify the limitations of the size of the baseline tree and decomposition.

Response: The reviewer is right, we should have exploited this concept further. Following the reviewer’s suggestion, we have made reference to these multi-scale aspects of our approach in the Introduction, Methods (Section 2(b)) and Discussion. We also discussed about the optimality in the solutions, in the Discussion section (text in blue to address critique from another reviewer).

3. In the section of the experiment (d), more description about vascular network creation is needed. I suppose that the authors applied a staged algorithm to create the vascular network consistent with anatomical knowledge. How long is the radius of the baseline network? How did the authors decompose the computational region? In discussion, the authors described that “In this process, the first generations of segmental vessels were automatically placed in the renal core, and from there, the interlobar arteries, the arcuate vessels, and finally, the interlobular arteries,” however, the section of the experiment (d) does not explicitly show how the authors achieved this process. I suppose that these multiscale vasculatures seem to be generated mainly by the staged algorithm, and the proposed parallel algorithm was exclusively used in

microscale vasculatures formation. The main contribution of the proposed parallel algorithm in this multiscale vascular network formation should be clarified.

Response: We have expanded the explanation regarding the generation of the vascular tree in Section 3(e) in the new version of the manuscript, describing the different stages of the algorithm. The qualitative description of “the interlobar arteries, the arcuate vessels, and finally, the interlobular arteries” was only a description for the non-specialist. We have also outlined the description of the subdomains employed in the partition of the model, as seen in Figure 10.

4. The size of the figures is too small to read this paper in print. Even on the monitor, I cannot read the words in figures without magnifying the pdf to at least 400%.

Response: We have increased the font visibility in all figures in the new version of the manuscript.

5. Figure 4 in line 33 on page 5 and line 8 on page 6 may be Figure 1.

Response: Indeed, the cross-reference was wrongly placed. This has been corrected in the new version of the manuscript.

We thank the reviewers for their contribution to our manuscript.
With best regards,

The authors